# Observation of giant dipole moments of interlayer excitons via layer engineering

Jiasen Zhu [1,4], Ting Liang [1,4], Fuhuan Shen [1,4] ✉, Zefeng Chen[1,2] ✉ & Jianbin Xu [1,3] ✉

Interlayer excitons in van der Waals (vdW) heterostructures (HSs) have garnered significant attention due to their unique properties, including prolonged lifetimes and long-range transport. While extensive studies have been conducted on interlayer excitons in HSs composed of different monolayers, research on HSs formed by multilayer constituents remains limited, particularly regarding dipole moments, which play a crucial role in light-matter interactions. In this study, we investigate the dipole moments of interlayer excitons in multilayer $WS_2$ and InSe HSs using the quantum-confined Stark effect. Our findings reveal that the dipole moment increases monotonically with the number of layers in InSe or $WS_2$, reaching a maximum of 3.18 $e$ nm, which is the largest value reported to date. Consequently, the dipole-dipole interaction is enhanced with the increasing layer number, as demonstrated by excitation power-dependent measurements. Ab initio calculations further support our experimental results, indicating the delocalization of the excitonic wave function with increasing layer thickness. Our findings introduce a novel layer-engineered mechanism for tuning the dipole moments of interlayer excitons in vdW heterostructures, paving the way for manipulating many-body interactions in low-dimensional quantum systems.

Heterostructures (HSs), formed by vertically stacking two different types of two-dimensional (2D) materials, exhibit exotic optoelectronic and quantum properties that are unattainable in their individual constituents[1-3]. Unlike their intralayer counterparts, interlayer excitons (IXs) are generated when electrons and holes are spatially separated across different layers, resulting in fundamentally distinct behaviors[4-6]. For example, these spatially indirect excitons demonstrate prolonged lifetimes—an order of magnitude longer than those of intralayer excitons—and possess long-range transport properties[5]. This leads to intriguing phenomena such as Bose–Einstein condensation at elevated temperatures and low-threshold lasing[7]. Moreover, the out-of-plane dipole moment characteristic of IXs allows for significant tunability (over 100 meV) in their spectra through the application of external fields, a phenomenon known as the quantum-confined Stark effect[8].

Recently, IXs trapped in a Moiré potential created by a precise "magic" twist of the constituent layers have given rise to a multitude of quantum phenomena, including superconductivity and fractional quantum anomalous Hall states[9]. HSs provide a versatile platform for various fields, including nanoelectronics, condensed matter physics, and quantum optics, with properties that can be tailored through multiple degrees of freedom, such as material selection and stacking configurations.

In previous studies, the quantum behavior of IXs formed within hetero-bilayers has been extensively investigated, where electrons and holes are confined within their respective constituent monolayers, typically exhibiting type-II band alignment[3]. Recent research has expanded the focus from bilayer HSs to trilayer (or above) systems, unveiling new types of IXs. For instance, spatially trapped IXs have

[1]Department of Electronic Engineering and Materials Science and Technology Research Center, The Chinese University of Hong Kong, Hong Kong SAR, PR China. [2]School of Electronic Science and Engineering (School of Microelectronics), South China Normal University, Guangzhou, PR China. [3]Shenzhen Research Institute, The Chinese University of Hong Kong, Shenzhen, PR China. [4]These authors contributed equally: Jiasen Zhu, Ting Liang, Fuhuan Shen. ✉e-mail: fhshenbbd@gmail.com; zefengchen@m.scnu.edu.cn; jbxu@ee.cuhk.edu.hk

been observed by sandwiching a bilayer hBN between $MoSe_2$ and $WSe_2$ monolayers[10,11]. Additionally, quadrupolar excitons—such as those with one hole located in the central layer and two electrons residing in the top and bottom layers, respectively—have been identified in trilayers like $WSe_2/WS_2/WSe_2$ HS (as well as in $WSe_2/WS_2/WSe_2$ and $WS_2/WSe_2/WSe_2$ HSs)[12–16]. These quadrupolar excitons exhibit a field-dependent dipole moment and reduced dipole-dipole interaction. In these multilayer systems, the carriers (either electrons or holes) that form the IXs remain confined within their specific monolayers, meaning that the dipole moment of these IXs is constrained by the interlayer spacing between the constituent monolayers.

Recently, pioneering works have shown that by extending one constituent material from monolayer to multilayer, the photo-luminescence properties of IXs—such as resonance energy and valley lifetime—can be modified due to the renormalized band structures that arise with increasing layer number[17,18]. However, the layer-dependent behavior of the dipole moment of IXs in multilayer HSs has yet to be explored.

In this study, we systematically investigate the variation of interlayer dipole moments in multilayer HSs using the quantum-confined Stark effect. We observe an unusually large interlayer dipole moment in multilayer $WS_2/InSe$ system with the recorded value up to 3.18 $e$ nm, in strikingly contrast to the general believes that IXs are confined between adjacent layers. Through continuously varying the layer number of InSe from 3 layers (3L) to 6L, a monolithic increase of dipole moment of IX is observed, which is evidently manifested with the increased energy shift under the same applied field (Stark effects). Similar trend is also found when the layer number of $WS_2$ changed from 2L to 3L. The repulsive dipole-dipole interaction is enhanced with the larger dipole moment, which is unambiguously manifested in the power-dependent measurements. The ab-initio calculation results support our measured results, reveals the delocalization of carriers with the layer number which underpins the enhanced dipole moment of IXs. Our findings shed light on the layer-engineered dipole moment of IXs in multilayer HSs, showing the new paradigm for the tunable photoelectronic and quantum devices at low-dimension[19–26].

## Results

### Variation of dipole moment of IXs with different layer number of InSe

A representative dual-gate device is schematically illustrated in Fig. 1a, featuring a bottom gate (Au film) and a top gate that independently control the voltage applied to each, effectively eliminating the influence of static doping (see details in Supplementary Note S1). The HSs are formed using $WS_2$ and InSe, both protected by top and bottom hexagonal boron nitride (h-BN) layers. Following the strategy in

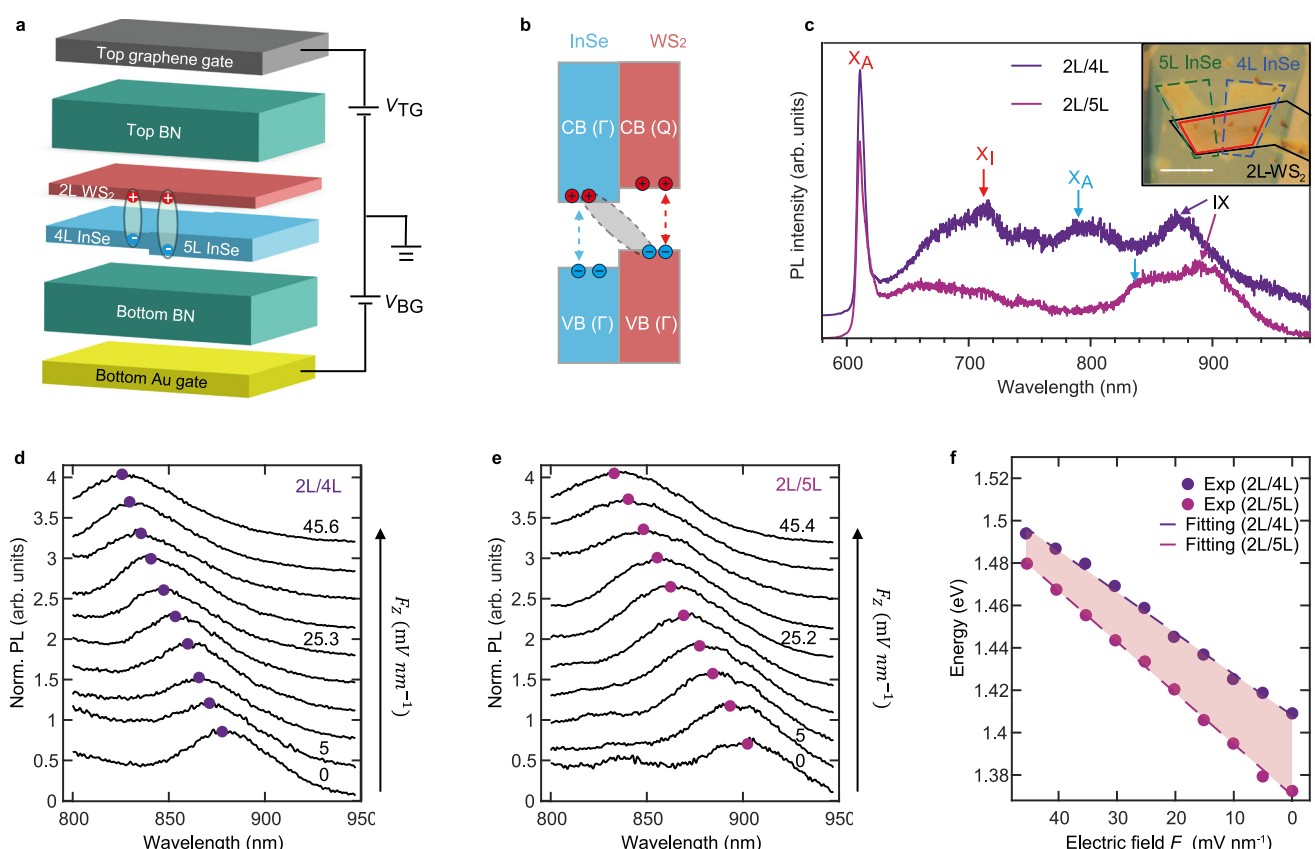

**Fig. 1 | The variation of IX properties with different layer number of InSe. a** A schematic illustration of a representative device applied in this work. The device consists of 2L $WS_2$ assembled with 4L and 5L InSe, with top (graphene) and bottom (gold) gate electrodes. **b** Schematic band structure of the $WS_2/InSe$ HS. Blue (Red) rectangles represent the band structures of the sole InSe ($WS_2$) layer. Blue (red) arrows indicate intralayer transitions in InSe ($WS_2$) layer. $X_A$: A exciton in 2L-$WS_2$. $X_I$: indirect transition in 2L-$WS_2$. The interlayer transition occurs between the valence band (providing holes) of $WS_2$ and the conduction band (providing electrons) of InSe at Γ point. **c** Measured PL spectra of the 2L/4L HSs (dark purple) and 2L/5L HSs (light purple) at 77 K. Inset: microscopic image of the $WS_2/InSe$ HS encapsulated with top and bottom h-BN. The HS is on a gold film (50 nm) serving as the bottom gate, with a graphene layer serving as the top gate. Individual 2D flakes are outlined for clarity: $WS_2$ (black dashed area), 4L InSe (blue dashed area), 5L InSe (green dashed area). Blue and red arrows indicate the corresponding peaks by pristine $WS_2$ and InSe layer. The scale bar is 10 μm. **d** Evolution of PL emission spectra of IX for the 2L/4L HS with the increasing field. **e** Same as (**d**) but for the 2L/5L HS. **f** Electric-field dependence of the PL peak positions of IXs extracted from (**d, e**).

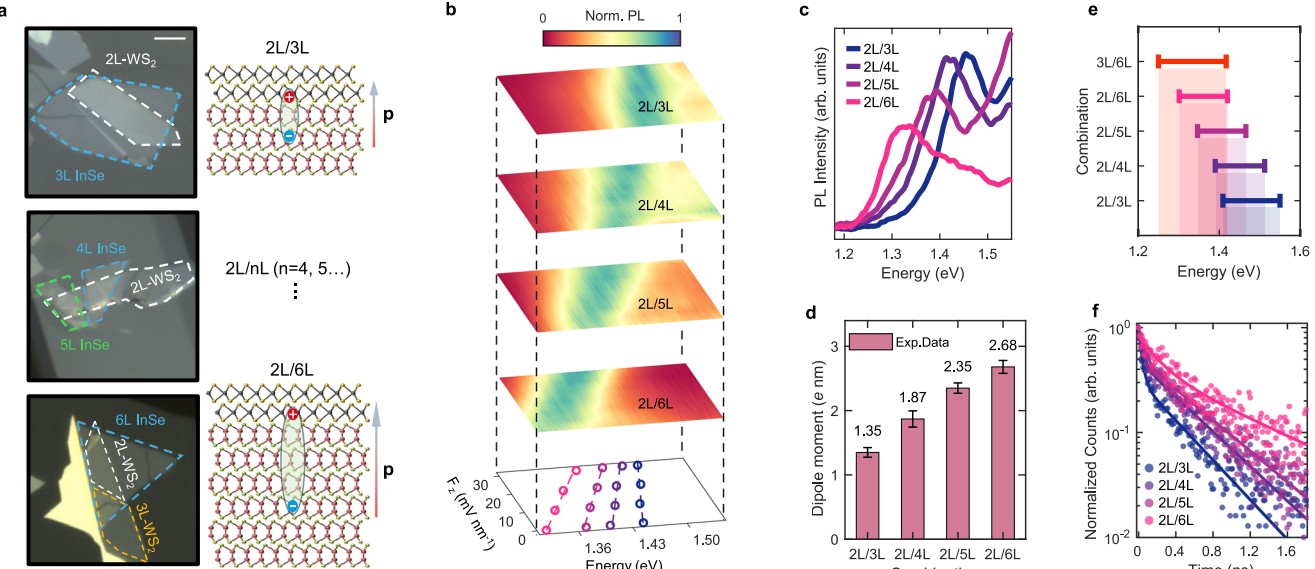

**Fig. 2 | Thickness-dependent properties of IX dipole in 2L-WS$_2$/nL-InSe HSs.**
**a** Microscope images of three multilayer HSs devices: device D1: 2L-WS$_2$/3L-InSe;
device D2: 2L-WS$_2$/4L-InSe and 2L-WS$_2$/5L-InSe; device D3: 2L-WS$_2$/6L-InSe. For
detailed images of individual materials and the stacking process, see Supplementary Fig. S4–6. The right inset schematically illustrates the increasing dipole
moment of IX with the increasing layer number of InSe. (Atoms: sulfur in gold,
tungsten in gray, indium in pink, and selenium in olive green.) **b** Pseudocolor map
of PL intensity as a function of applied electric field for HSs shown in (**a**). **c** PL
spectra of the 2 L/nL HSs (n from 3 to 6). **d** Calculated dipole moments of IXs for the
corresponding HSs in (**b**). The comparison of voltage and electric field conversion
is shown in Supplementary Fig. S8. The error bars arise from the linear fittings to the
Stark shifts. **e** The tunable range of resonance energy of IXs for different layer
combinations. **f** Measured PL lifetimes of IXs for different layer combinations: blue
dots (2L/3L), dark purple dots (2L/4L), light purple dots (2L/5L), pink dots (2L/6L).
The lines with corresponding colors denote their respective fitting curves,
respectively.

previous work[27], we select 2L-WS$_2$ and multilayer (from 3L to 6L) InSe
to achieve type-II band alignment at the Γ point (Fig. 1b). This alignment circumvents the momentum mismatch issues typically encountered by IXs at the K point, which require precise lattice matching of
the constituent materials. Consequently, IX transitions in the multilayer WS$_2$/InSe system are insensitive to the twisted angle between the
van der Waals layers, providing a viable platform for studying the
dipole behavior of multilayer HSs. To directly assess the influence of
layer number on the photoluminescence properties of IXs, we carefully
select monolithic InSe layers of varying thicknesses (specifically, 4L
and 5L) to combine with 2L-WS$_2$ (as shown schematically in Fig. 1a, with
the inset in Fig. 1c showing a microscopic image). This approach
ensures nearly identical experimental conditions for the device, aside
from InSe thickness. For clarity, we denote the HS as mL/nL, where m
represents the layer number of WS$_2$ and n represents the layer number
of InSe.

Figure 1c presents the photoluminescence (PL) spectra for the two
HS regions, i.e., 2L/4L and 2L/5L combinations. In addition to the PL
signatures from pristine WS$_2$ and InSe layers (with a direct comparison
of IXs to intralayer excitons from the constituent materials shown in
Supplementary Fig. S2), IXs from these two HS regions are observed in
the lower energy range of 850 nm to 900 nm, consistent with previously reported results[27]. The resonant energy of IXs exhibits a pronounced redshift as the thickness of the InSe layer increases, which is
attributed to band structure renormalization.

IXs in vdW HSs exhibit a static out-of-plane dipole moment due to
the spatial separation of electrons and holes across different layers.
With the external vertical field applied, the energy shift of dipolar IX
would be induced, i.e., $\Delta U = - \mathbf{p} \cdot \mathbf{E}_{HS}$, where $\mathbf{p} = e\mathbf{d}$ represents the
dipole moment (with $e$ being the electron charge and $\mathbf{d}$ the displacement between the electron and the hole which is also known as dipole
size). Utilizing this well-known Stark effects, the dipole moment of IX
can thus be extracted through field-dependent PL spectra. For the 2L/
4L HS region (Fig. 1d), the IX exhibits a blue shift of around 86 meV

from 1.404 eV to 1.490 eV with the electric field increased from 0 to
~45 mV/nm. In contrast, for the 2L/5L HS region (Fig. 1e), the IX exhibits
a blue shift of around 106 meV with the energy shift from 1.373 eV to
1.479 eV under almost the same variation of the electric field. The
difference is more evident in Fig. 1f where the evolution of peak
position of IXs are extracted from Fig. 1d and Fig. 1e, respectively. The
distinct slopes represent different dipole moments ($|\mathbf{p}| = \Delta U / |\mathbf{E}_{HS}|$),
which are calculated as $|\mathbf{p}| \approx 1.87$ $e$ nm (2L/4L) and $|\mathbf{p}| \approx 2.35$ $e$ nm
(2L/5L) separately. These extracted dipole moments significantly
exceed the previously reported values (0.5–0.8 $e$ nm)[28] for HSs composed of vdW monolayers, suggesting that the dipole size of the
multilayer HS systems remarkably surpasses the interlayer spacing
between constituent vdW layers.

### Layer-engineered out-of-plane dipole moment
To further investigate the dependence of increasing dipole moments
with the layer number of InSe, we fabricated additional 2L/3L and
2L/6L HSs (Fig. 2a). The layer numbers of InSe flakes were identified via
PL spectroscopy (see Supplementary Fig. S3). The IX shows an unambiguous and continuous redshift (Fig. 2c) with the increasing layer
number of InSe due to the band structure renormalization, which is
consistent with the previous work.

The evolution of IXs, with PL spectra normalized for each field
strength, is presented in Fig. 2b for different layer combinations. For all
HSs, the corresponding IX exhibits a nearly linear blue shift with
increasing vertical electric field. The rate of blue shift with field
strength (i.e., $\Delta U / |\mathbf{E}_{HS}|$) unambiguously increased with the number of
InSe layers, as indicated by the extracted peak positions obtained
through Lorentzian fitting (see Supplementary Figs. S9–S13) shown in
the bottom image of Fig. 2b. The calculated dipole moments of IXs
show a notable increase, ranging from ~1.35 $e$ nm for the 2L/3L combination to around 2.68 $e$ nm for the 2L/6L combination. In addition to
the HSs presented in the main text, we also fabricated a series of similar
HSs, which were fabricated using the same method to ensure

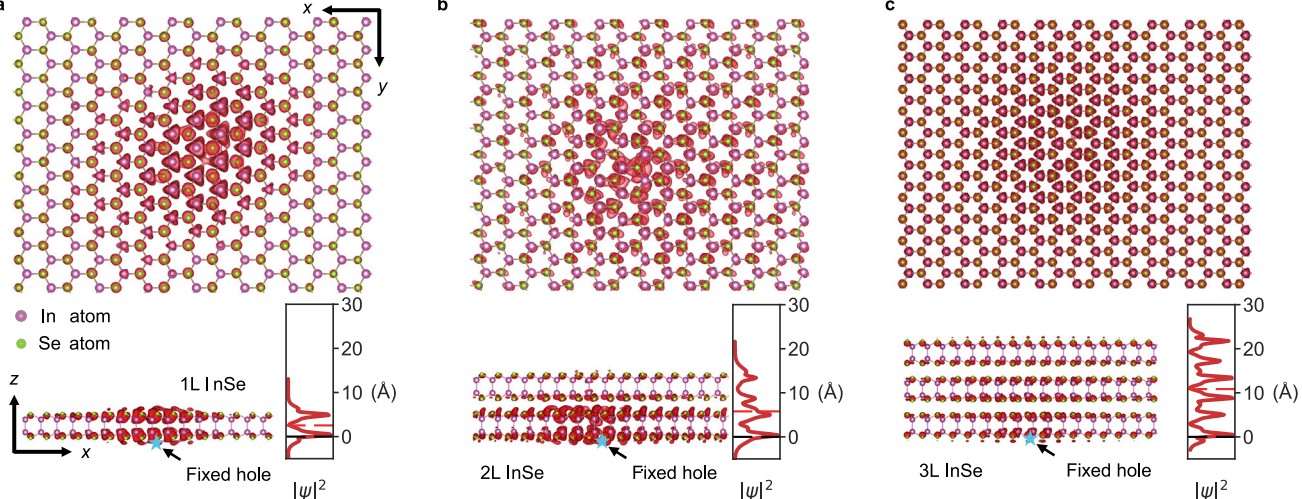

**Fig. 3 | Calculated hole wave function distribution for InSe with the increasing layer number. a–c** Top and side views of exciton wave functions for 1L-, 2L-, and 3L-InSe. The fixed hole (blue stars) position is indicated by the black arrows. The side image in each side view represents the exciton wave function density along the out-of-plane direction. Black dashed lines are hole position. Red dashed lines are the equivalent center position of the wave function.

consistency and reliability of the results. (see Supplementary Figs. S14–16) Additionally, Supplementary Figs. S18 and S19 illustrates the IX evolution under negative fields and larger positive fields. Under negative fields, the PL of IX demonstrates a clear near-linear redshift, while its intensity decreases dramatically, attributed to reduced electron-hole wave function overlap due to Coulomb forces (in contrast, PL intensity increases with positive fields). As the negative field strength further increases, the IX energy remains constant, likely due to a charging effect also noted in previous studies[28]. On the other hand, as the positive field continues to increase, the blue shift of IX begins to saturate. Band structure analyses suggest that at larger positive fields, the IX transition shifts from type-II band alignment at the Γ point to the InSe intralayer exciton transition, resulting in an unchanged peak position in the measured PL with further increases in positive field (Supplementary Fig. S17). Nevertheless, the linear Stark shift for all HSs is around 100 meV (Supplementary Fig. S19), ensuring the accuracy of the extracted dipole moments for the corresponding HSs. Through both the electric and layer engineering, the energy of IX can be continuously tuned across a broad range, as indicated in Fig. 2e and Supplementary Fig. S27.

It is anticipated that as the dipole size increases, the overlap of the electron-hole wave function is correspondingly reduced, leading to a prolonged lifetime for interlayer recombination. The lifetime measurements for different layer combinations, shown in Fig. 2f, unambiguously confirm this trend, with the lifetime increasing from 459 ps for the 2L/3L HS to 1069 ps for the 2L/6L HS. This significant increase in lifetime further supports the relationship between dipole size and recombination dynamics in these systems.

**First-principle calculation**
To further elucidate the behavior of IXs in multilayer WS₂/InSe HSs, we employed the GW-BSE (GW-Bethe-Salpeter Equation) method[29,30] to investigate their electronic structure and excitonic properties in detail (for the computational details, refer to Supplementary Note S2). However, due to the significant lattice mismatch between WS₂ and InSe, an especially large supercell composed of these two layers would be required for the full computation of the whole system. Instead, without loss generality, a simplified model is proposed to give the qualitative illustration of our observation in our experiment.

In this model, the hole is fixed at the bottom selenium (Se) atom of the InSe layer (as indicated by blue stars in the bottom panels of Fig. 3a–c) to simulate hole confinement within the WS₂ layer. This

approach mimics the experimental scenario where holes remain localized in WS₂, allowing us to study the electron behavior in InSe.

In Supplementary Fig. S23, we systematically compare the influence of fixing the electron (Supplementary Fig. S23a) versus the hole (Supplementary Fig. S23b) on the dipole moment calculation for WS₂/WSe₂, where their lattices exhibit a close match. The resulting dipole sizes from these two configurations are nearly identical. Furthermore, we calculated the hole wave function distributions by fully considering both 1L-WS₂/1L-WSe₂ and 1L-WS₂/2L-WSe₂ configurations, as well as by only considering the 1L-WSe₂ and 2L-WSe₂ configurations with the electron fixed at the edge of the WSe₂ layer (which aligns closely with our simplified model). The results show that the calculated hole wave function distributions are similar (as depicted in Supplementary Fig. S24). The primary difference lies in the dipole moment calculated when only considering WSe₂, which is significantly smaller than that derived from the full configurations due to the absence of interlayer spacing. However, the increase in dipole moment associated with the increasing layer number of WSe₂ remains consistent across both configurations. These analyses validate that, although the model simplifies the system by eliminating one of the constituent materials in the HS, the carrier wave function distributions and the layer-engineered dipole moment increase can still be accurately reflected by our simplified model.

Figure 3a–c present the top views (top panels) and side views (bottom panels) of the wave function distribution of electrons, with the hole fixed, for 1L-, 2L-, and 3L-InSe. Due to Coulomb attraction, the electron distribution is centered in the x-y plane (as observed in the top views). Moreover, there is a pronounced reduction in the probabilities of electron distribution as the layer number of InSe increases, attributed to Coulomb screening. The side view of the wave function distribution reflects the trend observed in our experiments. As shown in Fig. 3a–c, while the electron remains confined within the InSe layers where the hole is fixed at the bottom Se atom, its spatial extension systematically shifts toward upper layers with increasing InSe thickness.

By integrating the probability density ($|\psi|^2$) over the x-y plane, we quantitatively characterize the z-direction distribution (represented by the red curves adjacent to each side view), revealing that the excitonic wave function in 2L-InSe is strongly localized in the bottom layer nearest to the hole position, whereas in 3L-InSe it becomes nearly uniform across all layers, a difference likely originating from fundamental symmetry distinctions between even- and odd-layered InSe

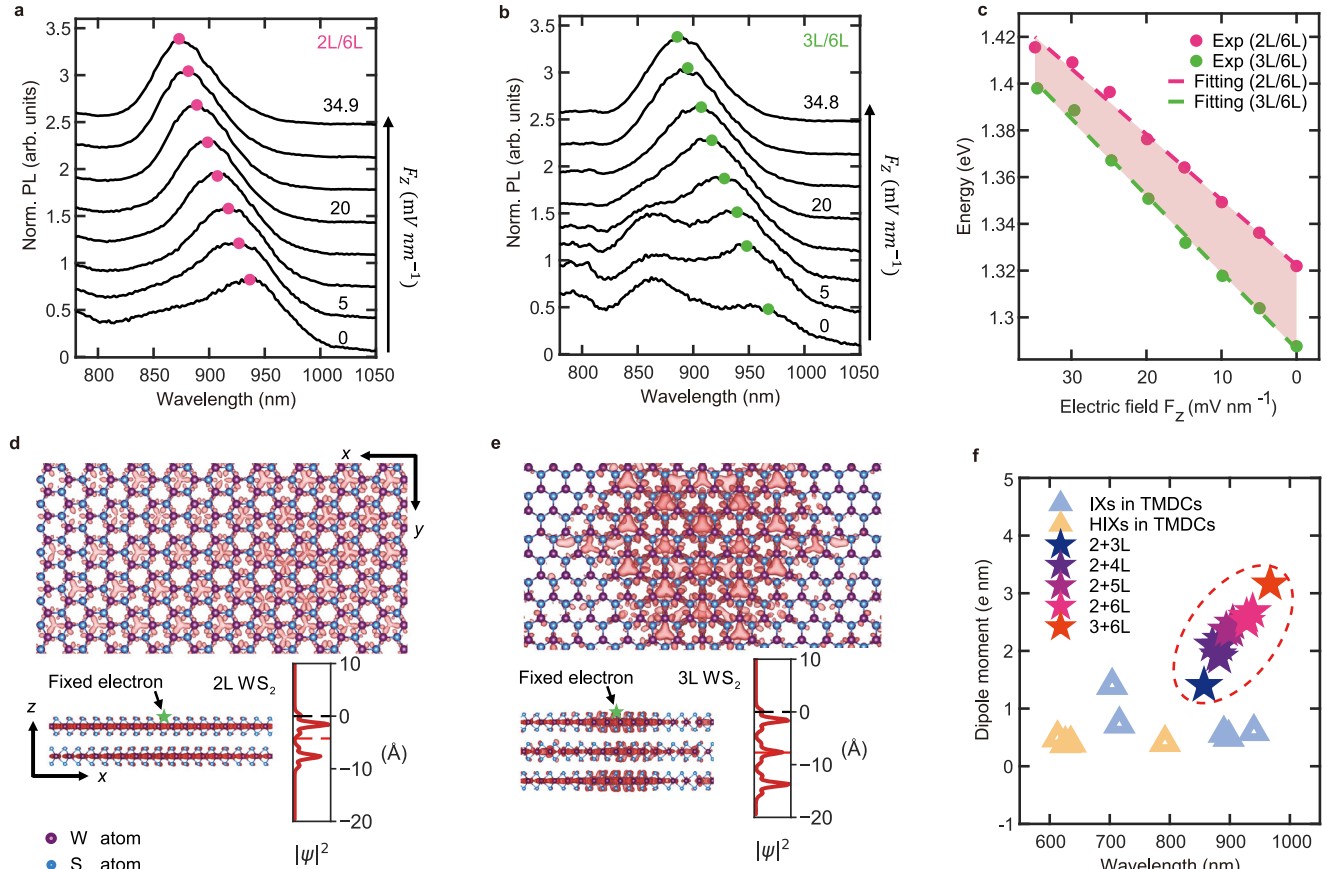

**Fig. 4 | Increasing IX dipole moment with the increasing WS₂ layer number.**
**a** Evolution of PL emission spectra of the 2L/6L HSs interface under an applied voltage. **b** Evolution of PL emission spectra of the 3L/6L HSs interface under the same applied voltage. **c** Electric field dependence of the interlayer exciton energies in 2L/6L and 3L/6L HSs. **d, e** Top and side views of exciton wave functions for 2L- and 3L-WS₂. The fixed electron position is indicated by the black arrow. The inset in each side view is the exciton wave function density along the out-of-plane direction.

Black dashed lines are electron positions. Red dashed lines are the equivalent center position of the wave function. **f** Comparison of performances of IXs in this work with those from other systems, including IXs in TMDCs and hybrid interlayer-intralayer excitons (hIXs) in TMDCs[8,16,17,31–35], obtained experimentally in mL/nLHSs. Theoretically, the dipole moments and working wavelength regime in mL/nL HSs can be further extended by choosing different thickness combinations.

systems. The effective Bohr radius is calculated by $r_B = \int \varphi^* r \varphi$ where $r$ is the relative distance between electron and fixed hole. The resultant calculated radius $r_B$ is 2.54 Å for 1L-InSe, 5.78 Å for 2L-InSe, and 10.99 Å for 3L-InSe, whose value is lower than the measured results in Fig. 2c (e.g., 13.5 Å for 2L/3L system) owing to the elimination of WS₂ lattice in the calculation.

Due to the limitations of our computational capabilities, we considered up to 3 layers of InSe. However, the trends observed in these calculations align closely with our experimental findings. The theoretical interlayer spacing increases by 3.24 Å from 1L to 2L and by 5.21 Å from 2L to 3L, which closely parallels the experimental increments of 5.2 Å (from 3L to 4L), 4.7 Å (from 4L to 5L), and 3.3 Å (from 5L to 6L). Additionally, the delocalization of electrons−indicated by the reduction of electron-hole wave function overlap with increasing layer number of InSe−supports the observed increase in dipole moment and is consistent with our lifetime measurement results. This coherence between theoretical predictions and experimental data underscores the robustness of our model and its relevance in understanding the behavior of excitons in multilayer InSe systems.

**Evolution of dipole moment with the layer number of WS₂**
In addition to varying the layer number of InSe, we also explored the impact of WS₂ thickness on the dipole moment. Two HSs−2L-WS₂/6L-InSe and 3L-WS₂/6L-InSe−were fabricated, all encapsulated in h-BN

layers (as shown in the microscopic image in Fig. 2a). Electric field-dependent PL measurements (ranging from 0 to 34.9 mV/nm) demonstrate characteristic Stark shifts. The 2L/6L configuration exhibits a 93.6 meV blue shift (from -1.3219 to 1.4155 eV, as shown in Fig. 4a), while the 3L/6L structure shows a larger 110.3 meV shift (from around 1.2876 to 1.3979 eV, depicted in Fig. 4b) under nearly identical electric fields. Linear regression analysis of these shifts (illustrated in Fig. 4c) yields dipole moments of 2.68 $e$ nm (2L/6L) and 3.18 $e$ nm (3L/6L), respectively. Notably, the latter value represents, to our knowledge, the largest IX dipole moment ever reported in vdW HSs.

Employing a similar methodology as in our previous calculations (Fig. 3), where we fixed the electron position at the edge of WS₂ layer to model 6L InSe while varying WS₂ thickness, generate excitonic wave functions whose spatial distributions (Fig. 4d, e) provide microscopic insight into the enhanced dipole moments observed in thicker WS₂ configurations. The side-view depiction reveals that, the wave function of holes in WS₂ exhibits relatively uniform interlayer delocalization. Specifically, in 2L-WS₂, the hole wave function shows slight concentration in the layer proximal to the electron, while in 3L-WS₂, it becomes more evenly distributed across all layers. Z-direction distribution shows the effective center of the hole wave function (red dashed line) shifts away from the fixed electron position (black dashed line), with the effective Bohr radius of 4.24 Å (2L-WS₂) and 7.7145 Å (3L-WS₂)−qualitatively consistent with our experimental observation of a 5 Å increment in the 2L/6L to 3L/6L HSs.

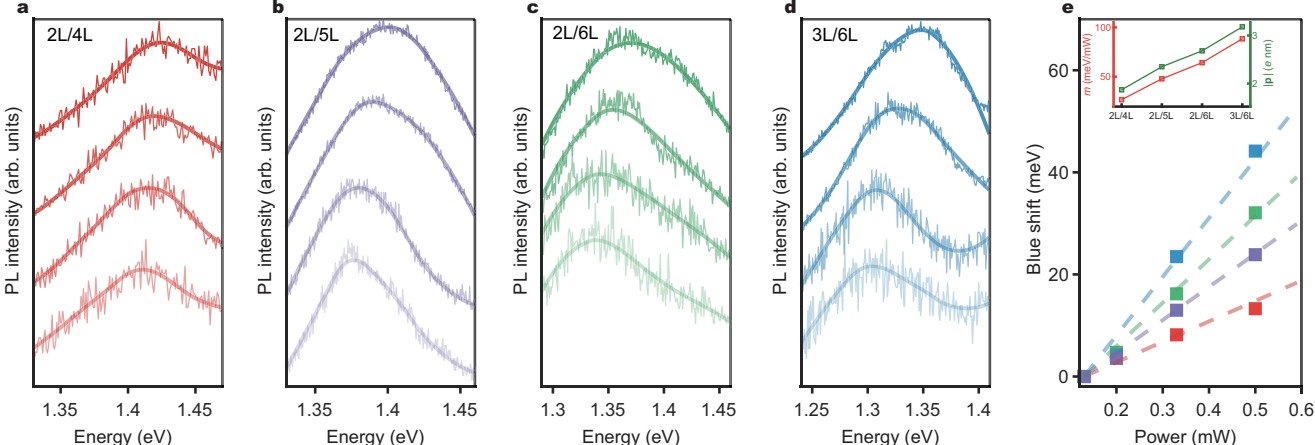

**Fig. 5 | Power-dependent behavior of IXs for different layer combinations.** **a–d** Evolution of PL spectra of IX for 2L/4L, 2L/5L, 2L/6L, and 3L/6L. The increasing excitation powers (from bottom to top) for **a–d** are 0.13 mW, 0.2 mW, 0.325 mW, and 0.5 mW, respectively. **e** The blue shift versus excitation power for various layer combinations. Inset: comparison of the blue shift increasing rate (m) and corresponding dipole moments.

This variation aligns qualitatively with the experimentally observed changes in the electric dipole moment, yet exhibits slight discrepancies in magnitude. Models that fix hole locations may not fully account for dynamic behavior. For instance, in the actual heterostructure, the holes are not completely fixed in the $WS_2$ layer but can migrate within and between layers. Additionally, the interfacial effects of the heterostructure, such as defects or doping at the contact surfaces, are often neglected. These interfacial effects can lead to nonradiative recombination and enhance the dielectric screening effect, thereby reducing the exciton binding energy.

Our study of IXs in multilayer HSs reveals that these systems exhibit exceptional electrical tunability due to giant Stark effects. The interlayer excitonic transitions in multilayer HSs cover a broad spectral range from ~1.25 eV to ~1.55 eV (Supplementary Fig. S27), offering promising opportunities for developing advanced photonic, nonlinear optical, and quantum devices. Importantly, the excitonic dipole moment can be precisely tuned by varying the thickness of constituent materials, enabling controlled manipulation of dipole-dipole interactions.

### Enhanced dipole-dipole interaction with the increasing layer number

The existence of an out-of-plane dipole moment of IX leads to mutual interactions, which manifest as a blue shift with increasing IX density. As the dipole moment increases, these mutual interactions are expected to be enhanced. Figure 5a–d show a clear blue shift with increasing excitation power (the excited exciton density is linearly proportional to the excitation power) across different layer combinations. Previous work[26] indicates that the energy shift can be estimated using the formula:

$$\Delta E_{XX} = n_0 \frac{e^2 |\mathbf{d}|}{\varepsilon_0 \varepsilon_{HS}} \tag{1}$$

where $\Delta E_{XX}$ represents the energy shift due to the dipole-dipole interaction, $n_0$ is the exciton density, $\varepsilon_0$ is the vacuum permittivity, $\varepsilon_{HS}$ is the relative permittivity of the heterostructure, $e$ is the electron charge, and $\mathbf{d}$ is the dipole size of the IX.

Figure 5e illustrates the IX position as a function of excitation power for different layer combinations, indicating that the energy shift rate ($m = \Delta E_{XX}/P$ where $P$ represents excitation power) increases with the layer number of $WS_2$ or InSe. The inset in Fig. 5e compares the corresponding dipole moments with the energy shift rate, revealing a strong correlation. These results clearly demonstrate that the dipole-dipole repulsion interaction is significantly enhanced due to the increased dipole moment engineered by the layer number.

Last but not least, while the layer configuration up to the 3L/6L combination exhibits the largest dipole moment observed in our experiments, it is clear that the upper limits have not yet been reached (see Supplementary Fig. S25). Two factors could lead to the disappearance of IXs with increasing layer number. First, band structure alignment plays a crucial role. As the layer number increases, the type-II band structures (illustrated in Fig. 1b) may become invalid, resulting in interlayer-intralayer exciton transitions. Second, binding energy is another critical factor. As the dipole size increases, the binding energy correspondingly decreases due to reduced Coulomb attraction. When the binding energy falls below the thermal energy of the environment, IXs can no longer be observed. Detailed analyses can be found in Supplementary Note S3, which provides upper limits for the dipole moment: ~6.28 e nm due to the limitations by band structure alignment and about 3.85 e nm due to the limitations by binding energy at 77 K. The latter estimation can be increased with the decreased temperature.

## Discussion

In summary, our work demonstrates the layer-engineered dipole moment of IXs in HSs. As the layer numbers of InSe and $WS_2$ increase, the dipole moment shows a significant increase, reaching a maximum of 3.18 e nm, the largest value reported to date. Theoretical calculations reveal that this increasing dipole moment results from the delocalization of electrons and holes, aligning perfectly with the experimental trends observed. The enhanced dipole moment leads to stronger dipole-dipole interactions, which are reflected in the power-dependent PL for various layer combinations. Furthermore, through layer engineering and electrical control (Stark effects), we achieve continuous modulation of IX resonance from ~1.25 eV to ~1.54 eV. Our work paves the way for manipulating the dipole moment of IXs, which could advance the study of many-body quantum phenomena and enable the development of tunable, broadband, low-dimensional devices.

It is well known that, compared to the negligible oscillator strength (ranging from $10^{-3}$ to $10^{-4}$ of that of intralayer excitons) in HSs, IXs in homostructures, such as bilayer $MoS_2$, exhibit finite oscillator strength, which can be detected in reflection or absorption measurements. Recently, the every-other-layer exciton observed in trilayer $WSe_2$ has demonstrated a large dipole moment of ~1.4 e nm. However, these multilayer homostructures are indirect bandgap semiconductors, resulting in relatively low quantum efficiency for IX transitions, which limits their applications in exciton condensation or lasing. Moreover, although a larger dipole moment is observed for every-other-layer IXs, protected by lattice symmetry, increasing the layer number does not lead to further enhancements in the dipole moment.

## Methods

### Device fabrications

Gold (50 nm) and titanium (5 nm, adhesion layer) films were sequentially deposited on a silicon substrate by electron-beam evaporation at a deposition rate of 1 Å/s. Few-layer graphene, bilayer $WS_2$, multilayer InSe, and hexagonal boron nitride (h-BN) flakes were mechanically exfoliated onto polydimethylsiloxane (PDMS) stamps. The HSs were assembled using a polypropylene carbonate (PPC)-assisted dry transfer method with precise alignment, followed by dissolution of the PPC stamp in acetone. The h-BN thickness was characterized by atomic force microscopy (Bruker Dimension Icon).

### Optical measurements

All optical measurements were performed using a bright-field/dark-field confocal microscope. The photoluminescence (PL) spectrum was measured using a HR550 Horiba Jobin Yvon spectrometer. PL spectra were measured in an optical cryostat (Linkam) with a 532 nm laser excitation. A 50× objective lens with a numerical aperture of 0.5 was used. For time-resolved photoluminescence (TRPL) measurements, a second harmonic 1064-nm (532 nm) Ti:sapphire femtosecond laser beam (Coherent Chameleon Ultra II) was used to illuminate the sample. The laser had a repetition rate of 80 MHz. The signal was collected using a 50× objective lens (LCPLAN N, NA = 0.65, OLYMPUS, JAPAN). The final signal was sent to a single-photon detector (MPD PDM Series), and the output was recorded using a Picoharp 300 for time-correlated single-photon counting (TCSPC) measurements.

## Data availability

All data that support the plots within this paper and other findings of this study are available from the corresponding author upon request.

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

## Acknowledgements

The work is in part supported by the National Key R&D Project from the Ministry of Science and Technology of China (No. 2022YFA1203100), National Natural Science Foundation of China (62274114), Research Grants Council of Hong Kong, particularly, via Grant AoE/P-701/20, CUHK Group Research Scheme, RGC Postdoctoral Fellowship, CUHK Postdoctoral Fellowship, Basic and Applied Basic Research Foundation of Guangdong Province (No.2023B1S1S120049). T.L. thanks Dr. Xin Wu for valuable help with GW-BSE calculations. J.Z. and F.S. thank Dr. Kai Feng for the meaningful discussion, and thank Prof. Dangyuan Lei and Mr. Shuaiyu Jin for the help of lifetime measurements.

## Author contributions

J.Z. and F.S. conceived the idea. F.S., Z.C., and J.X. guided the work. J.Z. fabricated the samples. J.Z., F.S., and Z.C. performed the optical experiments. T.L. performed the DFT and GW-BSE calculations. J.Z., F.S., J.X., and Z.C. made the data analysis. All the authors contributed to the data analyses and paper writing.

## Competing interests

The authors declare no competing interests
