## [Transparent Peer Review file · Nature Communications]

Observation of Giant Dipole Moments of Interlayer Excitons via Layer Engineering

Corresponding Author: Professor Jianbin Xu

Version 0:

Reviewer comments:

Reviewer #1

(Remarks to the Author)

Zhu et al. investigate interlayer excitons (IXs) in van der Waals heterostructures (HSs) composed of multilayer WS₂ and InSe. Using photoluminescence (PL) spectroscopy under applied electric fields, the authors demonstrate that the dipole moment of IXs increases monotonically with the layer number of either InSe or WS₂, reaching a record value of 3.18 e-nm. This behavior is attributed to the delocalization of electron wavefunctions in thicker layers, supported by GW-BSE calculations. The topic is of interest for light-matter interaction however the result is presented in a rather confusing way and, in my opinion, does not justify enough improvement over previous IX or hIX papers. I cannot suggest publishing in high impact journal like nat comms, a revised version will be more suitable for more specialized journal like npj 2d mater appl, Comm Phys or PRB.

1. The layer dependent Gamma transition was previous studied systematically (ref. 25). The authors claim their main finding on top of that is the electric field dependence, i.e. very large dipole moment of over previous works. However, I find this a hard sell. The point of pursuing large dipole moment is: 1) enabling large spectra tunability in Stark shift 2) introducing large dipolar repulsive/attractive interaction to realize large IX diffusion or collective behavior like quadrupolar excitons, respectively. Authors fail to introduce advancement in these two crucial fields, for 1) their Stark shift saturate after 0.03 V/nm, resulting in similar spectra tunability over previous papers. From the way the authors present their data I cannot get clear picture at higher field, as Fig4 reaches the conventional field strength but present very few line cuts at high field. For 2) I do not see any exciton density dependent characterization or many-body effect.

2. Following previous comment, despite the claim of having 'record high' dipole moment to meet the eye, all the main characterizations like dipole extraction are confined within 0.03 V/nm. This does not make fair comparison with other system showing continuous linear Stark shift towards the 0.3 V/nm or larger field range. In other words, this system limits the useful field range one order of magnitude lower than conventional vdW devices. The reason could be due to the intrinsic Gamma transition for this largely layer-engineered structure. As the effective e-h separation increases the intrinsic dipolar moment, it always risk reducing exciton oscillator strength or binding energy, which results in shift saturation at very low field. Some PL linecuts in Supp info also show quite dim emission which suggest weak oscillator strength. In comparison the hIX or every-other-layer IX e.g. (ref. 11) may show a better balance with both enhanced dipole moment as well as sufficient oscillator strength due to interlayer tunneling, which stabilize the field dependence of dipole. In this work when it goes to few layer-few layer structure the interlayer tunneling is simply not strong enough, hence the very limited range of Stark shift. In a word the represented data does not show enough scientific merits over previous IX system, thus I cannot agree with the technical and application potential claimed by the authors.

3. Some typos are found in line 49 it should be 'every-other-layer exciton' rather than every-other exciton.

Reviewer #2

(Remarks to the Author)

The submitted manuscript, "Observation of Giant Dipole Moment of Interlayer Excitons in Multilayer Heterostructure", reports a systematic study on interlayer excitons in multilayer WS₂/InSe heterostructures. By varying the thickness of constituent layers, the authors demonstrate a prominent increase in exciton dipole moment, reaching up to 3.18 e-nm — the largest value reported to date in van der Waals heterostructures. These findings are substantiated by quantum-confined Stark effect

measurements and supported by ab initio simulations. The manuscript offers valuable insight into dipole engineering via layer number control, thus extending the design toolkit for next-generation infrared and excitonic optoelectronic devices. Notably, the authors reveal that the interlayer excitonic wavefunction becomes increasingly delocalized as the thickness of InSe or WS₂ increases. This delocalization underlies the enhancement of the vertical dipole length beyond the conventional vdW gap distance. The study combines careful device design, spectroscopic analysis, and theoretical calculations to provide a comprehensive picture of dipole evolution in such multilayer systems. Given the novelty and potential impact of the results, I find this manuscript to be of strong interest to readers of Nature Communications. However, I recommend the authors address the following concerns before it is suitable for publication:

Major Concerns:

1. Lack of Reversed Field Data:

While the manuscript provides compelling evidence of linear Stark shifts under positive gate bias, it lacks data under reversed (negative) field direction. Since a hallmark of dipolar excitons is a symmetric energy response to field reversal, this omission weakens the claim. The authors should either present reversed-field data or clearly explain why it is not available, and whether any asymmetry arises from built-in fields or device structure.

2. WS₂ Emission Behavior in Heterostructures (Fig. 1d):

In Figure 1d, the WS₂ exciton peak within the heterostructure region exhibits a shift in both position and lineshape compared to pristine WS₂. This suggests possible charge transfer or interface interactions. The authors should clarify whether these effects influence the interpretation of interlayer exciton resonance and whether they affect the extraction of Stark shifts.

3. Color Similarity in Figure 1d:

The PL spectra of 4L-InSe and 5L-InSe in Figure 1d are rendered in colors that are difficult to distinguish (cyan vs aqua green). This reduces the figure's clarity and accessibility. The authors are encouraged to choose more distinguishable colors or adopt different line styles to improve visual clarity.

4. Clarify Theoretical Assumptions on Carrier Fixation

The theoretical modeling assumes either the electron or the hole is fixed at a certain atomic layer (e.g., bottom Se in InSe) to simplify exciton wavefunction calculations. However, the justification for this approximation is not well explained in the main text. Given the importance of wavefunction asymmetry in determining the dipole moment, the authors should clarify why this assumption is valid, what physical behavior it aims to reproduce, and how it affects the accuracy of the extracted dipole moments.

5. Clarify Treatment of Interlayer Coupling in Simulations:

The exciton dipole enhancement is attributed to wavefunction delocalization across layers, but the manuscript does not clearly explain how interlayer coupling is treated in the simulations. The authors should briefly clarify whether and how interlayer hybridization is included when calculating the electron wavefunction distribution, as this directly impacts the extracted dipole trend.

6. Discuss the Upper Limit of Dipole Moment

While the authors report record-high excitonic dipole moments, the manuscript lacks a discussion on the theoretical and experimental upper bounds of dipole length in such multilayer heterostructures. It would be valuable to briefly address what limits the maximum achievable dipole moment—e.g., exciton binding energy, dielectric screening, or wavefunction delocalization saturation—and whether further increases in layer number are expected to continue enlarging the dipole moment or lead to diminishing returns.

Overall Assessment:

This manuscript reports a compelling study of tunable interlayer excitons with giant dipole moments in multilayer WS₂/InSe heterostructures. The results are novel, experimentally robust, and supported by theory. Clarifying the modeling assumptions, addressing dipole symmetry under reversed fields, and discussing the upper bound of dipole strength would further improve the work. With minor revisions, the paper is suitable for publication in Nature communications.

Reviewer #3

(Remarks to the Author)

The manuscript titled "Observation of Giant Dipole Moment of Interlayer Excitons in Multilayer Heterostructure" presents a systematic investigation of dipole moments in WS₂/InSe heterostructures with varying layer numbers.

The study demonstrates that the dipole moment increases monotonically with the number of InSe or WS₂ layers, reaching a maximum value of 3.18 e·nm in the 3L WS₂/6L InSe configuration.

The experimental data are well-presented, detailed, and informative, though the findings are not entirely unexpected.

While the work is solid, I recommend acceptance after revisions to address the following points:

1. The manuscript examines 2L WS₂/nL InSe (n=3–6), but 1L- and 2L-InSe cases are absent. What is the rationale for omitting these configurations? Additionally, the WS₂ thickness is fixed at 2L in most cases. What would be the expected behavior for 1L WS₂/nL InSe

heterostructures? A brief discussion and experiment data on this would strengthen the study's comprehensiveness.

2. The results suggest that thicker heterostructures yield larger dipole moments, with the maximum observed in 3L WS₂/6L InSe. Does this trend continue beyond 6L InSe or 3L WS₂? Is there a theoretical or experimental upper limit to the dipole moment in such systems? A discussion on this would provide deeper insight.

3. While PL spectroscopy effectively probes interlayer exciton energies, sometime reflectance measurements could offer additional insights—particularly in distinguishing intralayer exciton contributions and hybrid interlayer exciton states. Including such data would allow direct comparison with prior studies and enhance the manuscript's robustness.

4. The manuscript contains numerous spelling errors (e.g., References [5, 9, 13, 14, 19, 30, 31...]), suggesting insufficient attention to detail. Journal name formatting is inconsistent (e.g., "Nature Communications", "Nat Commun", "Nat Photonics"). The references should follow a uniform style (preferably abbreviated journal names, e.g., Nat. Commun., Nat. Photon.).

Best regards,

Version 1:

Reviewer comments:

Reviewer #1

(Remarks to the Author)

I thank the authors for the reply. I appreciate that the authors clarify the interlayer to intralayer character transition of this specific heterostructure. This does not remove my concern about the impact of achieving large dipole but only within very small field, i.e. only limited ΔE . Nevertheless I acknowledge the novelty of this feature in its own context compared to K valley exciton in MX₂. The added power dependent PL makes sense but not unexpected. The authors stated that their 2L/3L device can not stand a field beyond 0.08 V/nm. which seems like a very low threshold. A second device or some clarification would be encouraged.

Note that the authors reply that they layer engineered the IX dipole in heterostructure for the first time, but I disagree with such claim. Fig 6c in supp info of Nat. Mater. 19, 630–636 (2020) showed clear dipole tuning of K valley IX with various component thickness. Similarly see also Fig S7 and S8 in Ref.19 indicating similar effect on K valley IX. Various papers with heterobilayer separated by BN spacer in principle is also a type of dipole-layer engineering.

Overall in my opinion the authors did not provide exceptional critical new results in their revision, but I acknowledge the findings provide new insight for dipole engineering and interlayer-intralayer transition in the context of Γ -related excitons. Authors reply to all the technical questions from other referees in details. I can suggest acceptance of the paper after addressing the comments above.

Reviewer #2

(Remarks to the Author)

The authors have successfully addressed my concerns. The paper is in good shape now and can be accepted.

Reviewer #3

(Remarks to the Author)

The authors addressed all of my questions thoroughly. I recommend that this manuscript be published in Nature Communications.

Response letter for the manuscript (NCOMMS-25-35571) entitled “Observation of Giant Dipole Moments of Interlayer Excitons via Layer Engineering”

Reviewer 1

Zhu et al. investigate interlayer excitons (IXs) in van der Waals heterostructures (HSs) composed of multilayer WS₂ and InSe. Using photoluminescence (PL) spectroscopy under applied electric fields, the authors demonstrate that the dipole moment of IXs increases monotonically with the layer number of either InSe or WS₂, reaching a record value of 3.18 e·nm. This behavior is attributed to the delocalization of electron wavefunctions in thicker layers, supported by GW-BSE calculations. The topic is of interest for light-matter interaction however the result is presented in a rather confusing way and, in my opinion, does not justify enough improvement over previous IX or hIX papers. I cannot suggest publishing in high impact journal like nat comms, a revised version will be more suitable for more specialized journal like npj 2d mater appl, Comm Phys or PRB.

Response: We thank reviewer for the feedback and appreciate the time took to review our work. The major contributions of our work to the fields of IXs can be summarized in the following:

Unveiling the Layer-Engineered Dipole Moment of IXs in Heterostructures: Our study is, to the best of our knowledge, the first systematic investigation of the layer-dependent behavior of the dipole moment of IXs in heterostructures. We find that the dipole moment of IXs can be monotonically tuned with the increasing number of layers, achieving a record-high dipole moment of approximately 3.18 enm. In previous studies, either the dipole moment of IXs in heterostructures or the excitons in every other layer of homotrilayers remained fixed. Our work paves the way for exploring tunable low-dimensional devices.

Revealing the Delocalization of Carrier Wavefunctions via Ab Initio Methods: The mechanism of delocalization of electron or hole wavefunctions increases with the number of layers, as demonstrated by our ab initio calculations. This finding aligns perfectly with the observed increase in dipole moment and prolonged lifetime as the layer number increases.

Observing Enhanced Dipole-Dipole Interactions with Increasing Layer Number: Following the reviewers'

suggestions, our power-dependent measurements indicate that dipole-dipole interactions are significantly enhanced with the increasing dipole moment. This enhancement may have important implications for the fields of Bose–Einstein condensation and high-temperature superconductivity [1, 2].

Our work provides substantial experimental and theoretical evidence to support these findings. In response to the reviewers' questions, we have provided detailed point-to-point responses below.

Comment 1. The layer dependent Gamma transition was previous studied systematically (ref. 25). The authors claim their main finding on top of that is the electric field dependence, i.e. very large dipole moment of over previous works. However, I find this a hard sell. The point of pursuing large dipole moment is: 1) enabling large spectra tunability in Stark shift 2) introducing large dipolar repulsive/attractive interaction to realize large IX diffusion or collective behavior like quadrupolar excitons, respectively. Authors fail to introduce advancement in these two crucial fields, for 1) their Stark shift saturate after 0.03 V/nm, resulting in similar spectra tunability over previous papers. From the way the authors present their data I cannot get clear picture at higher field, as Fig. 4 reaches the conventional field strength but present very few line cuts at high field. For 2). I do not see any exciton density dependent characterization or many-body effect.

Our response: We thank the reviewer for this insightful comment. The major contribution of our work is the unveiling of the layer-engineered dipole moment of interlayer excitons (IXs) in heterostructures, which enables the observation of a giant dipole moment (~ 3.18 eÅ) in the experiments. While previous study [3] has explored layer-dependent behaviors of IXs (PL peak position) at the Γ point, which is attributed to the renormalized band structure as the number of layers increases, our findings present distinct advancements: (1) We are the first to reveal the layer-engineered dipole moment of IXs in heterostructure experimentally. (2) We achieve a record-high dipole moment in heterostructures. (3) We realize enhanced dipole-dipole interactions through layer engineering (following the reviewer's suggestions).

For the reviewer's first concern on spectra tunability by Stark effect:

According to the dipole model, i.e., $\Delta E = -\mathbf{p} \cdot \mathbf{E}$ where \mathbf{p} is the dipole moment and \mathbf{E} is the external electric field, the Stark shift (ΔE) will be larger for a larger dipole moment under the same applied electric field.

This relationship is unambiguously confirmed in our experiment. However, a larger dipole moment does not necessarily lead to a broader spectral range that can be tuned by the electric field. As we analyze in the following, we find that the extent of energy shift induced by the external field is actually determined by the band alignment of the constituent materials, rather than the dipole moment.

As shown in **Figure R1a**, the multilayer InSe and WS₂ form a type-II band alignment, with the IX transition occurring between the conduction band minimum (CBM) of InSe at the Γ point and the valence band maximum (VBM) of WS₂, also at the Γ point. **Figure R1b** presents a simplified band structure that corresponds to state 2 shown in **Figure R2**.

Figure R1. (a) Band structures of InSe and WS₂ showing type-II band alignment. (b) Simplified band alignment for IX transition.

With a positive external field applied (transitioning from state 2 to state 3 in **Figure R2**), the CBM of InSe and the VBM of WS₂ move further apart in the energy domain, resulting in a blue shift of the IX energy. As the positive field is further increased (transitioning to state 4), an interlayer-to-intralayer exciton transition occurs, where the VBM of WS₂ becomes lower than the VBM of InSe. In this scenario (state 4), the type-II band alignment for the IX transition is no longer valid. With an even greater increase in the field (state 5), the intralayer exciton emission from InSe, which cannot be tuned by the electric field, dominates, resulting in a constant PL energy. Conversely, under a negative (reversed) electric field (i.e., transitioning from state 2 to state 1 in **Figure R2**), the energy experiences a corresponding red shift.

The band structure analyses align perfectly with the experimental results when the electric field extends

beyond the region of linear Stark effects. As shown in **Figure R3a**, for the 2L/3L heterostructure, the significant energy difference between the IX and the InSe intralayer exciton allows for continuous tuning of the energy (blue shift) as the field increases beyond 80 mV/nm. However, a higher field cannot be applied due to limitations in the voltage that the device can withstand. In contrast, for the 2L/4L, 2L/6L, and 3L/6L heterostructures (**Figures R3b-d**), the energy blue shift saturates as the IX energy approaches the energy of the InSe intralayer exciton (indicated by the green arrows). To achieve a larger tuning range for the IX, alternative material combinations with a greater band contrast could be explored.

Nevertheless, since the primary focus of our work is to investigate the layer-dependent behavior of the dipole moment of IXs, we adopted Stark effect measurements as our characterization method. The linear Stark effect, resulting in an energy shift of around 100 meV, is sufficient for extracting the dipole moment of the corresponding IX.

Figure R2. Schematic illustration of the evolution of band structures of multilayer InSe/WS₂ heterostructure with external field.

Figure R3. Evolution of IX emission with electric field for different combinations of heterostructures: 2L/3L (e), 2L/4L (f), 2L/6L (g), and 3L/6L (h). (a-d) Corresponding PL spectra at zero electric field.

Revision: Figure R2 and Figure R3 were added as **Supplementary Figure 17** and **Figure S19** in the revised supplementary materials.

The discussion of the saturation of blue shift with increasing electric field was added on the revised manuscript:

“On the other hand, as the positive field continues to increase, the blue shift of IX begins to saturate. Band structure analyses suggest that at larger positive fields, the IX transition shifts from type-II band alignment at the Γ point to the InSe intralayer exciton transition, resulting in an unchanged peak position in the measured PL with further increases in positive field (**Supplementary Figure S17**). Nevertheless, the linear Stark shift for all heterostructures is around 100 meV (**Supplementary Figure S19**), ensuring the accuracy of the extracted dipole moments for the corresponding HSs.”

For the reviewer’s second concern regarding enhanced many-body interactions with increasing dipole moments:

We have supplemented our analysis with power-dependent measurements across various heterostructures. As shown in **Figures R4a-d**, the IX PL peak clearly exhibits a blue shift with increasing excitation power, which

we attribute to enhanced dipole-dipole repulsion [4]. Moreover, these interactions strengthen with the increasing dipole moment. **Figure R4e** illustrates the extracted blue shift (ΔE) relative to excitation power for different heterostructure combinations, each displaying distinct rates of increase (defined as $m = \Delta E/P$ where P represents the excitation power). According to previous work [5], m is proportional to the dipole moment of IXs. The inset provides a direct comparison of the corresponding dipole moments and m , revealing a significant consistency.

Figure R4. Power-dependent PL spectra for 2L/4L (a), 2L/5L (b), 2L/6L (c), and 3L/6L (d) heterostructures. The increasing excitation powers (from bottom to top) for (a-d) are 0.13 mW, 0.2 mW, 0.325 mW, and 0.5 mW, respectively. (e) Energy shift with the increasing excitation power for 2L/4L (red), 2L/6L (green), and 3L/6L (blue) heterostructures. Inset: comparison of blue shifts (with respect to the power) and dipole moment for different layer combinations.

Revision: Figure R4 was added as the **Figure 5** in the revised manuscript. The related power-dependent analyses is attached:

“**Figure 5e** illustrates the IX position as a function of excitation power for different layer combinations, indicating that the energy shift rates ($m = \Delta E_{XX}/P$ where P represents excitation power) increase with the layer number of WS₂ or InSe. The inset in Figure 5e compares the corresponding dipole moments with the energy shift rate m , revealing a strong correlation. These results clearly demonstrate that the dipole-dipole repulsion interaction is significantly enhanced due to the increased dipole moment engineered by the layer number.”

Comment 2. Following previous comment, despite the claim of having ‘record high’ dipole moment to meet the eye, all the main characterizations like dipole extraction are confined within 0.03 V/nm. This does not make fair comparison with other system showing continuous linear Stark shift towards the 0.3 V/nm or larger field range. In other words, this system limits the useful field range one order of magnitude lower than conventional vdW devices. The reason could be due to the intrinsic Gamma transition for this largely layer-engineered structure. As the effective e-h separation increases the intrinsic dipolar moment, it always risk reducing exciton oscillator strength or binding energy, which results in shift saturation at very low field. Some PL linecuts in Supp info also show quite dim emission which suggest weak oscillator strength. In comparison the hIX or every-other-layer IX e.g. (ref. 11) may show a better balance with both enhanced dipole moment as well as sufficient oscillator strength due to interlayer tunneling, which stabilize the field dependence of dipole. In this work when it goes to few layer-few layer structure the interlayer tunneling is simply not strong enough, hence the very limited range of Stark shift. In a word the represented data does not show enough scientific merits over previous IX system, thus I cannot agree with the technical and application potential claimed by the authors.

Response: We thank the reviewer for raising this question. As mentioned in our response to the previous comment, the limits of the Stark shift are due to the band alignment of the constituent materials and the maximum field that the heterostructure can withstand. We achieved a energy shift range of approximately 100 meV for each heterostructure configuration (**Figure 2e** in the main text). Furthermore, by combining electric and layer engineering, we can achieve continuous broad band tunability for a large spectral range (from ~1.25 eV to ~1.55 eV, as shown in **Figure 2e** in the revised manuscript and **Supplementary Figure S27** in the revised Supplementary materials).

Followings are the specific points to reviewer’s questions:

1. Decrease of the PL intensity with the electric field.

The reduction in PL under electric field (as shown in **Figure R5a** where the 2L/3L heterostructure is adopted for an representative example) is attributed to the Coulomb force exerted by the external field. As schematically depicted in **Figure R5c**, a positive field exerts an attractive Coulomb force on the IX dipole, enhancing the overlap of the wavefunctions of the electron and hole, leading to the increased IX PL. Conversely, a negative

field results in a significant reduction in PL at relatively large negative values. This phenomenon occurs not only for at- Γ IXs but also for at-K IXs. For instance, the field-dependent PL of the WSe₂/WS₂ heterostructure [4] exhibits a close results to ours, as shown in Figure **R5b**.

Figure R5. (a) Same image as **Figure 3e**. (b) Cited image from the Figure 2a in the main text of **ref [3]**. (c) Schematic illustration of PL variation with the field.

2. Comparison of IXs Formed in Heterostructures and Homostructures

The formation of IXs in heterostructures and homostructures is fundamentally different. In heterostructures, the electron-hole separation results in an inherently negligible oscillator strength (approximately 10^{-3} to 10^{-4} smaller than that of intralayer excitons) [6]. Consequently, only PL measurements are used for IX characterization in heterostructures. Despite the small oscillator strength, carriers from different layers can tunnel to the VBM and CBM for the IX transition. Although an increased layer number generally reduces the binding energy and electron-hole wavefunction overlap, the pronounced electric-field-dependent PL observed due to Stark effects (Figures R3 and R4) evidences the existence of IXs in all heterostructure configurations presented in our manuscript.

In contrast, IXs in homostructures (e.g., IXs in MoS₂ homobilayers [7] or every-other-layer excitons in trilayer or >3L WSe₂ [8]) arise from the admixture of intralayer excitons and thus possess a much larger oscillator strength, allowing for measurement via reflection or absorption. However, the results from these

studies have the following constraints compared to our work:

1. The bilayer or trilayer (or thicker) homostructures are indirect bandgap semiconductors, where IX transitions are of low efficiency, leading to negligible PL emission (**Figure R6c**). This limitation impacts potential applications, including exciton condensation and lasing.
2. Although a larger dipole moment (~ 1.4 e nm) is observed for every-other-layer IXs, protected by lattice symmetry, increasing the layer number does not yield further enhancements in the dipole moment. That is, unlike layer-engineered IXs in our work, the every-other-layer IXs are of the fixed dipole moment.

Figure R6. Evolution of the differential reflectance spectra (**a**, cited from the **Figure 2a** in the ref [6]) and photoluminescence (**c**, cited from the **extended data Figure 2** in the ref [6]) of the every-other-layer exciton with the electric field. (**b**) Schematic illustration of IXs in heterostructure (left) and homostructure (right).

Revision: Figure R5 was added as **Supplementary Figure S20** in the revised supplementary materials. The comparison of our work with every-other-layer IXs in homostructures was added in the discussion in the revised manuscript. The related sentences are attached for reference:

“It is well known that, compared to the negligible oscillator strength (ranging from 10^{-3} to 10^{-4} of that of intralayer excitons) in heterostructures, IXs in homostructures, such as bilayer MoS_2 , exhibit finite oscillator

strength, which can be detected in reflection or absorption measurements. Recently, the every-other-layer exciton observed in trilayer WSe₂ has demonstrated a large dipole moment of approximately 1.4 enm. However, these multilayer homostructures are indirect bandgap semiconductors, resulting in relatively low quantum efficiency for IX transitions, which limits their applications in exciton condensation or lasing. Moreover, although a larger dipole moment is observed for every-other-layer IXs, protected by lattice symmetry, increasing the layer number does not lead to further enhancements in the dipole moment.”

Comment 3. Some typos are found in line 49 it should be ‘every-other-layer exciton’ rather than every-other exciton.

Response: We thank reviewer for pointing out the typos and we fixed all of them in the revised manuscript.

Reviewer 2

The submitted manuscript, "Observation of Giant Dipole Moment of Interlayer Excitons in Multilayer Heterostructure", reports a systematic study on interlayer excitons in multilayer WS₂/InSe heterostructures. By varying the thickness of constituent layers, the authors demonstrate a prominent increase in exciton dipole moment, reaching up to 3.18 e·nm — the largest value reported to date in van der Waals heterostructures. These findings are substantiated by quantum-confined Stark effect measurements and supported by ab initio simulations. The manuscript offers valuable insight into dipole engineering via layer number control, thus extending the design toolkit for next-generation infrared and excitonic optoelectronic devices.

Notably, the authors reveal that the interlayer excitonic wavefunction becomes increasingly delocalized as the thickness of InSe or WS₂ increases. This delocalization underlies the enhancement of the vertical dipole length beyond the conventional vdW gap distance. The study combines careful device design, spectroscopic analysis, and theoretical calculations to provide a comprehensive picture of dipole evolution in such multilayer systems.

Given the novelty and potential impact of the results, I find this manuscript to be of strong interest to readers of Nature Communications. However, I recommend the authors address the following concerns before it is suitable for publication:

Response: We thank the reviewer for the recognition and high evaluation of our work. The point-to-point response is detailed in the following.

Comment 1. Lack of Reversed Field Data:

While the manuscript provides compelling evidence of linear Stark shifts under positive gate bias, it lacks data under reversed (negative) field direction. Since a hallmark of dipolar excitons is a symmetric energy response to field reversal, this omission weakens the claim. The authors should either present reversed-field data or clearly explain why it is not available, and whether any asymmetry arises from built-in fields or device structure.

Response: We thank the reviewer for this meaningful suggestion. Following the reviewer’s recommendations, we measured the Stark shifts under both positive and negative fields, which are presented in **Figure R7**.

Figure R7a-d shows the evolution of PL with positive and negative fields for different heterostructure combinations, while **Figure R7e-h** presents the corresponding peak positions for field-dependent IXs. For all heterostructures shown in **Figure R7a-d**, the intensity of the IX PL decreases due to reduced electron-hole wavefunction overlap caused by the Coulomb force, as schematically illustrated in **Figure R5c**.

Specifically, for the 2L/3L combination, the PL peak positions exhibit an (almost) linear relationship with the negative field, displaying the same slope as observed under positive fields. A slight deviation from linearity is noted when the negative field strength exceeds 20 mV/nm. For the other combinations (2L/4L, 2L/6L, and 3L/6L), the linear Stark shift is initially observed at low field strengths (less than 20 mV/nm), after which the PL peak positions saturate (i.e., remain unchanged) with increasing negative field strength. For the 3L/6L combination, the PL intensity becomes too weak under larger negative fields (>20 mV/nm), making it difficult to extract the corresponding peak positions. This deviation from linear Stark shifts under large negative fields has been reported in other studies (e.g., in Ref [9]) which is attributed to charging effects. Despite the observation of energy shift saturation at negative fields, the linear Stark shift exceeding 100 meV across the (small) negative to positive field range provides a sufficient region to extract the dipole moments of IXs for the various heterostructure combinations.

Figure R7. Evolution of IX PL with the electric field for 2L/3L (a), 2L/4L (b), 2L/6L (c), and 3L/6L (d) heterostructures where mL/nL represents that heterostructure is composed of mL-WS₂ and nL-InSe. (e-h) Corresponding IX peak positions extracted from (a-d). Red line is the guide for the eye for the linear dispersion

of IX with electric field.

Revisions: Figure R7 was added as **Supplementary Figure S18** in the revised supplementary materials.

The analyses on negative field was added in the revised manuscript with the following sentences:

“Additionally, **Supplementary Figure S18** and **S19** illustrates the IX evolution under negative fields and larger positive fields. Under negative fields, the PL of IX demonstrates a clear near-linear redshift, while its intensity decreases dramatically, attributed to reduced electron-hole wave function overlap due to Coulomb forces (in contrast, PL intensity increases with positive fields). As the negative field strength further increases, the IX energy remains constant, likely due to a charging effect also noted in previous studies. On the other hand, as the positive field continues to increase, the blue shift of IX begins to saturate. Band structure analyses suggest that at larger positive fields, the IX transition shifts from type-II band alignment at the Γ point to the InSe intralayer exciton transition, resulting in an unchanged peak position in the measured PL with further increases in positive field. Nevertheless, the linear Stark shift for all heterostructures is around 100 meV (**Supplementary Figure S19**), ensuring the accuracy of the extracted dipole moments for the corresponding HSs.”

Comment 2. WS₂ Emission Behavior in Heterostructures (Fig. 1d):

In Figure 1d, the WS₂ exciton peak within the heterostructure region exhibits a shift in both position and lineshape compared to pristine WS₂. This suggests possible charge transfer or interface interactions. The authors should clarify whether these effects influence the interpretation of interlayer exciton resonance and whether they affect the extraction of Stark shifts.

Response: We thank the reviewer for highlighting this question. The energy shift of the WS₂ exciton between pristine 2L WS₂ and the heterostructure is attributed to the different temperatures during measurements (300 K for pristine 2L WS₂ and 80 K for the heterostructure). We apologize for the misleading information conveyed by this image. Additionally, we noticed that the lineshape of the PL spectrum near the WS₂ exciton in the original Figure 1d exhibited some peaks, which we attribute to instrument artifacts. By carefully adjusting the instrument setup, we have mitigated these artifacts.

Figure R8b-c shows the remeasured PL spectra from the 2L/4L and 2L/5L heterostructures, along with the corresponding PL spectra from their constituent WS₂ and InSe layers. As illustrated in **Figure R8b**, the 2L-WS₂ exhibits an A exciton emission (X_A , ~620 nm) and an indirect bandgap emission (X_{indirect} , ~720 nm), while the 4L-InSe shows an intralayer exciton emission around 800 nm (**Figure R8a** depicts their respective transition processes). After the formation of the heterostructure, the PL of both WS₂ and InSe is quenched due to charge transfer. For the 2L/4L heterostructure, the characteristic peaks from the constituent layers remain, with only slight energy shifts relative to their original positions. This minor shift can be attributed to changes in the dielectric environment or small strains during the transfer process, as observed in other studies [10]. In the case of the 2L/5L heterostructure (**Figure R8c**), similar conditions apply, but the X_{indirect} emission is completely quenched or obscured by the background of the IX spectrum.

Figure R8. (a) Band structure of InSe/WS₂ heterostructure. (b) Comparison of the PL spectra from 2L/4L heterostructure (blue curve), 2L-WS₂ (orange curve), and 4L-InSe (yellow curve). (c) Comparison of the PL spectra from 2L/5L heterostructure (purple curve), 2L-WS₂ (orange curve), and 5L-InSe (yellow curve).

Revision: We added **Figure R8** as **Supplementary Figure S2** in the revised supplementary materials. In addition, in the revised manuscript the PL spectra of heterostructures (**Figure 1c** in the revised manuscript) were replaced by the remeasured PL shown in **Figure R8**.

Comment 3. Color Similarity in Figure 1d:

The PL spectra of 4L-InSe and 5L-InSe in Figure 1d are rendered in colors that are difficult to distinguish

(cyan vs aqua green). This reduces the figure's clarity and accessibility. The authors are encouraged to choose more distinguishable colors or adopt different line styles to improve visual clarity.

Response: We appreciate the valuable suggestion from the reviewer. We have carefully adjusted the corresponding colors (**Figure 1c** in the revised manuscript).

Comment 4. Clarify Theoretical Assumptions on Carrier Fixation

The theoretical modeling assumes either the electron or the hole is fixed at a certain atomic layer (e.g., bottom Se in InSe) to simplify exciton wavefunction calculations. However, the justification for this approximation is not well explained in the main text. Given the importance of wavefunction asymmetry in determining the dipole moment, the authors should clarify why this assumption is valid, what physical behavior it aims to reproduce, and how it affects the accuracy of the extracted dipole moments.

Response: We thank the reviewer for this insightful question for about the simplified model we adopted in the calculations.

First, we clarify the rationale for fixing either the electron or the hole to a specific atomic layer. The exciton wavefunction is intrinsically six-dimensional, and directly visualizing such a function in real space is extremely difficult, if not impossible. Fixing the position of the hole (or the electron) effectively selects a particular two-dimensional slice of this six-dimensional space. This slice represents the spatial probability distribution of the electron (or the hole) when the hole (or the electron) is “pinned” at a specific position R_0 . The result is a three-dimensional function that can be directly visualized in real space. This approach provides the most straightforward and practical means of understanding the spatial distribution of excitons, their degree of localization, and the relative motion between electrons and holes. Such methodology is well established in the literature (see, e.g., [11-15]). Here, the 1L-WS₂/1L-WSe₂ is used as a representative example. By fixing the electron (hole) in 1L-WS₂ (WSe₂), we can calculate the hole wavefunction distribution in 1L-WSe₂ (1L-WS₂) correspondingly, as indicated in **Figure R9a** (**Figure R9b**). These two independent calculations yield nearly equivalent dipole sizes, with $d=4.53 \text{ \AA}$ for the case in **Figure R9a** and $d=4.6 \text{ \AA}$ for the case in **Figure R9b**.

Next, we discuss the implications and validity of this simplified model, and demonstrate that it can accurately capture the experimental observations. Due to the large lattice mismatch between InSe and WS₂, directly visualizing the exciton wave function of their heterostructure while fully accounting for the lattice mismatch is computationally intractable. Therefore, we adopt a simplified model in which either the electron or the hole in one material is fixed, and the spatial distribution of the other carrier within the same material is calculated (i.e., **Figure 3** and **Figure 4d, e** in the main text).

To validate the effectiveness of this approach, we consider multilayer WSe₂/WS₂ heterostructures, where the lattice constants are very similar, making it feasible to model the full heterostructure explicitly. **Figure R10a** and **R10b** compare the hole wave function distribution in monolayer WSe₂ obtained from the complete 1L-WSe₂/WS₂ heterostructure model (**Figure R10a**) and from the monolayer WSe₂-only calculation (**Figure R10b**). The two results are strikingly similar, thereby supporting the validity of our simplified model. Moreover, the effective dipole size using $r_B = \int \varphi^* r \varphi$ (r is the displacement with respect to the fixed electron) can be calculated as 4.6 Å (which is very close to the value reported in the previous results [9, 16]) for **Figure R10a** and 1.73 Å for **Figure R10b**. Because in a real case, the electron only resides within the WS₂ layer, the consideration in **Figure R10a** is thus close to real results while the simplified model in **Figure R10b** results in a large deviation.

We further calculate the 2L-WSe₂/1L-WS₂ configuration with fully considering the 2L-WSe₂/1L-WS₂ heterostructure (**Figure R10c**) and without considering 1L-WS₂ (**Figure R10d**). For both cases, the hole wavefunction distributions within 2L-WSe₂ are close, with the wavefunction distributed in both layers of WSe₂. The dipole size is thus calculated for these cases, giving the value of 7.66 Å (**Figure R10c**) and 4.37 Å (**Figure R10d**). Though a similar offset of calculated dipole size for these two cases is found, the increased dipole size with the increasing layer are is found to be similar. That is, for **Figure R10a** and **Figure R10c** where the fully heterostructure configuration is considered, the dipole size is increased by 3.06 Å (7.66 Å - 4.74 Å), which is very close to the dipole size change 2.64 Å (4.37 - 1.73 Å) for **Figure R10b** and **Figure R10d**.

As a result, through the above analysis, we can summarize the applicability and limitations of the simplified model for calculation in heterostructure:

1. Fixing either the electron or hole in the calculation of IXs in heterostructures gives a close value of dipole

moments.

2. The electron and hole wavefunction distributions by this simplified method provide a decent agreement with those calculated by the full heterostructure configuration, reflecting the delocalization of electrons or holes with the increasing layer number of constituent material.
3. The calculated dipole size for this simplified method shows the deviation from the results calculated by the full configuration, owing to the direct placement of electron or hole at the edge atom of one of the constituent materials.
4. Though the deviation of dipole size is found, the increased dipole size with the layer number shows a decent consistency with the results by considering the full configuration. As a result, the simplified model can convincingly reflect the trend of the increasing dipole moment with the increasing layer number, which is the major finding for our work.

Figure R9. (a) Hole wavefunction distribution within 1L-WS₂ for 1L-WS₂/1L-WSe₂ heterostructure. The electron is fixed at the WSe₂ layer. (b) Electron wavefunction distribution within 1L-WSe₂ for 1L-WS₂/1L-WSe₂ heterostructure. The hole is fixed at the WS₂ layer.

Figure R10. (a) The distributions of hole wavefunction in 1L-WSe₂ layer when it forms a heterostructure with 1L-WS₂ (a equivalent electron is fixed at the edge of WS₂). (b) The same as (a) but there is no 1L-WS₂ and the fixed electron is at the edge Se atom of WSe₂. (c-d) Same as (a-b) but for the 2L-WSe₂ system.

Revisions: Figure R9 and R10 were added as Supplementary Figure S23 and Figure S24 in the revised supplementary materials.

The discussion on validity and limitations was added in the revised manuscript with the following related sentences:

“In Supplementary Figure S23, we systematically compare the influence of fixing the electron (Supplementary Figure S23a) versus the hole (Supplementary Figure S23b) on the dipole moment calculation for WS₂/WSe₂, where their lattices exhibit a strong match. The resulting dipole sizes from these two configurations are nearly identical. Furthermore, we calculated the hole wave function distributions by fully considering both 1L-WS₂/1L-WSe₂ and 1L-WS₂/2L-WSe₂ configurations, as well as by only considering the 1L-WSe₂ and 2L-WSe₂ configurations with the electron fixed at the edge of the WSe₂ layer (which aligns closely with our simplified model). The results show that the calculated hole wave function distributions are similar (as depicted in Supplementary Figure S24). The primary difference lies in the dipole moment calculated when only considering WSe₂, which is significantly smaller than that derived from the full configurations due to the absence of interlayer spacing. However, the increase in dipole moment associated with the increasing layer number of WSe₂ remains consistent across both configurations. These analyses validate that, although the model simplifies the system by eliminating one of the constituent materials in the HS, the carrier wave function

distributions and the layer-engineered dipole moment increase can still be accurately reflected by our simplified model.”

Comment 5. Clarify Treatment of Interlayer Coupling in Simulations:

The exciton dipole enhancement is attributed to wavefunction delocalization across layers, but the manuscript does not clearly explain how interlayer coupling is treated in the simulations. The authors should briefly clarify whether and how interlayer hybridization is included when calculating the electron wavefunction distribution, as this directly impacts the extracted dipole trend.

Response: We thank the reviewer for this insightful and critical comment. We agree that a clearer explanation of how interlayer coupling is treated in our simulations is essential and should be added to the manuscript.

In our first-principles calculations (DFT and GW-BSE), interlayer coupling and hybridization are physical results intrinsically embedded within the computational framework, rather than manually introduced parameters.

When calculating the exciton wavefunctions in multilayer InSe (or WS₂), the Hamiltonian inherently includes all intra-layer and inter-layer hopping terms. Therefore, the distribution of the electron (hole) wavefunction is a result of interlayer hybridization and Coulomb attraction. Consequently, the distribution of the electron (hole) wavefunction is a result of the combined effects of interlayer hybridization and Coulomb attraction. In other words, the delocalized wavefunction features we obtained (as shown in the **Fig. 3** and **Fig. 4d-e** of the revised manuscript) are direct evidence of the existence of interlayer coupling.

Specifically:

1. **DFT calculation** self-consistently determines the electronic ground state of the system by solving the Kohn-Sham equations, which incorporate the potential from all atomic nuclei and the Coulomb and exchange-correlation potentials from all electrons. The resulting wavefunctions, if delocalized across layers, are a direct manifestation of interlayer coupling and hybridization. The use of Grimme's D3 correction ([cite: J. Chem. Phys. 2010, 132, 154104]) ensured the physical accuracy of the interlayer distances, providing a reliable foundation for this calculation.
2. The subsequent **GW calculation** performs quasiparticle corrections based on the wavefunctions and

electron density provided by the DFT step. It fully inherits the interlayer coupling and hybridization information from the DFT wavefunctions, modifying the energy levels without altering the spatial delocalization characteristics of the wavefunctions.

3. Finally, the **BSE calculation** uses the delocalized single-particle states from GW as a basis and solves the electron-hole interaction. The resulting exciton wavefunction naturally spans multiple layers, successfully capturing the trend of increasing dipole moment with layer number, which aligns qualitatively with our experimental results.

Revision: We added the related discussion on first-principle calculations in the revised supplementary materials:

"In our first-principle calculations (DFT and GW-BSE), interlayer coupling and hybridization are physical results intrinsically embedded within the computational framework, rather than manually introduced parameters. When calculating the exciton wavefunctions in multilayer InSe/WS₂, the Hamiltonian inherently includes all intra-layer and inter-layer hopping terms. Therefore, the distribution of the electron/hole wavefunction is a result of interlayer hybridization and Coulomb attraction. Furthermore, we employed the D3 correction ([cite: J. Chem. Phys. 2010, 132, 154104]) to ensure the physical accuracy of the interlayer distances, providing a reliable foundation for the calculation."

Comment 6. Discuss the Upper Limit of Dipole Moment

While the authors report record-high excitonic dipole moments, the manuscript lacks a discussion on the theoretical and experimental upper bounds of dipole length in such multilayer heterostructures. It would be valuable to briefly address what limits the maximum achievable dipole moment—e.g., exciton binding energy, dielectric screening, or wavefunction delocalization saturation—and whether further increases in layer number are expected to continue enlarging the dipole moment or lead to diminishing returns.

Response: We sincerely appreciate the reviewer for this insightful suggestion, which has facilitated a deeper discussion on interlayer excitons in multilayer heterostructures. As shown in **Figure R11**, the increase in dipole moment with the rising number of InSe or WS₂ layers is not saturated. There are primarily two factors that will

limit the maximum dipole moment we can achieve:

1. Disruption of Type-II Alignment with Increasing InSe and WS₂ Layers

As shown in **Figure R12a**, the energy difference between the VBM of InSe and that of WS₂ decreases from 0.39 eV to 0.18 eV as the number of InSe layers increases from 2L to 5L (according to Ref. [3]). Experimental data (**Figure R12b**, directly cited from **Figure 3a** in the reference [npj 2D Mater Appl 8, 12 (2024)]) indicates that this energy difference approaches zero as the number of InSe layers approaches 18, rendering the type-II alignment invalid. Consequently, we can roughly estimate the upper limit of the dipole moment at 18L InSe as $p_{largest} = (2.68 \text{ enm}) + (0.3 \text{ enm} \times 12) = 6.28 \text{ enm}$. Here, 2.68 enm is the dipole moment for the 2L/6L heterostructure, and we estimate that each additional layer increases the total dipole moment by 0.3 enm (based on the increase from 2L/5L to 2L/6L).

A similar analysis can be applied to WS₂. As indicated in **Figure R12c**, the CBM of InSe and WS₂ approaches each other with an increasing number of WS₂ layers. However, the variation in the CBM of WS₂ with layer number is smaller (as seen in Peak I of **Figure R12d**, which is directly cited from **Figure 3c** in [Sci Rep 3, 1608 (2013)]) compared to the trend observed in InSe. Increasing the WS₂ layers to bulk would result in a CBM of WS₂ that is very close to that of InSe, rather than surpassing it, which would compete with interlayer exciton transitions. However, due to the lack of further band structure data for various layers of WS₂, the estimation of the upper limits of the dipole moment with increasing WS₂ layers remains uncertain.

2. Binding energy:

The interaction between the electron and hole can be described by the Coulomb potential, which is proportional to $\frac{1}{\epsilon_r d}$ where ϵ_r is the relative permittivity of heterostructure (approximately treated as constant across different layer combinations) and d is the dipole size. Consequently, a larger dipole size results in weaker Coulomb attraction, thereby reducing the binding energy of the exciton.

In a previous report (supplementary materials in Ref. [3]), the binding energy for the 2L/3L heterostructure was estimated to be less than 20 meV. Given that the binding energy is proportional to $E_b \propto \frac{1}{d}$ and that $d=1.35$ nm for the 2L/3L combination, and considering that the binding energy must exceed the thermal energy of the

environment (i.e., $k_B T \approx 7$ meV for $T=77$ K), we estimate the maximum dipole size to be around 3.85 nm. However, the limits set by binding energy can increase with a decrease in temperature, as this leads to a reduction in the thermal energy from the environment.

Figure R11. The evolution of dipole moment with different layer combinations.

Figure R12. (a) Evolution of band structures of 2L-WS₂/NL-InSe heterostructure with the increasing layer number of InSe. (b) Measured PL spectra for InSe of various layer number (directly cited from Figure 3a in [*npj 2D Mater Appl* **8**, 12 (2024)].). (c) Evolution of band structures of NL-WS₂/2L-InSe heterostructure with the increasing layer number of WS₂. (d) The PL peak positions for WS₂ of various layer number (directly cited from

Figuer 3c in [*Sci Rep* **3**, 1608 (2013).].

Revisions: The related discussion on the upper limits of the dipole moment by layer-engineering in the revised manuscript is attached:

“Last but not least, while the layer configuration up to the 3L/6L combination exhibits the largest dipole moment observed in our experiments, it is clear that the upper limits have not yet been reached (see **Supplementary Figure S25**). Two factors could lead to the disappearance of interlayer excitons (IXs) with increasing layer number. First, band structure alignment plays a crucial role. As the layer number increases, the type-II band structures (illustrated in **Figure 1b**) may become invalid, resulting in interlayer-intralayer exciton transitions. Second, binding energy is another critical factor. As the dipole size increases, the binding energy correspondingly decreases due to reduced Coulomb attraction. When the binding energy falls below the thermal energy of the environment, IXs can no longer be observed. Detailed analyses can be found in **Supplementary Note S3**, which provides upper limits for the dipole moment: approximately 6.28 nm due to the limitations by band structure alignment and about 3.85 nm due to the limitations by binding energy at 77 K. The latter estimation can be increased with the decreased temperature.”

Overall Assessment:

This manuscript reports a compelling study of tunable interlayer excitons with giant dipole moments in multilayer WS₂/InSe heterostructures. The results are novel, experimentally robust, and supported by theory. Clarifying the modeling assumptions, addressing dipole symmetry under reversed fields, and discussing the upper bound of dipole strength would further improve the work. With minor revisions, the paper is suitable for publication in Nature communications.

Response: We thank reviewer again for the recognition of our work. Following the suggestions by the reviewer, the supplement of the reversed field and discussion on the upper bound of the dipole moment were added in the revised version of this work.

Reviewer #3 (Remarks to the Author):

The manuscript titled “Observation of Giant Dipole Moment of Interlayer Excitons in Multilayer Heterostructure” presents a systematic investigation of dipole moments in WS₂/InSe heterostructures with varying layer numbers. The study demonstrates that the dipole moment increases monotonically with the number of InSe or WS₂ layers, reaching a maximum value of 3.18 e·nm in the 3L WS₂/6L InSe configuration. The experimental data are well-presented, detailed, and informative, though the findings are not entirely unexpected. While the work is solid, I recommend acceptance after revisions to address the following points:

Response: We sincerely thank reviewer for the recognition of our work. We give the point-to-point response in the following.

Comment 1. The manuscript examines 2L WS₂/nL InSe (n=3–6), but 1L- and 2L-InSe cases are absent. What is the rationale for omitting these configurations?

Additionally, the WS₂ thickness is fixed at 2L in most cases. What would be the expected behavior for 1L WS₂/nL InSe heterostructures? A brief discussion and experiment data on this would strengthen the study’s comprehensiveness.

Response: We thank reviewer for this question on the layer number selection for the formation of the heterostructures. We follow the strategy in Ref [3] to form the type-II band alignment that supports interlayer transition at $k=0$ (i.e., at Γ point) in the Brillouin zone, and thus avoids momentum mismatch required, for instance, by the interlayer transition at K point in other heterostructure systems (such as 1L-MoS₂/1L-WSe₂ heterostructure shown in Ref [9]). For this purpose, the bilayer and thicker multilayer of WS₂ with its VBM at the Γ point, and multilayer InSe with its conduction-band minimum (CBM) also at Γ point are selected. This is schematically illustrated in the band structure alignment in **Figure R13a**.

Indeed, 2L-InSe also matches this goal (as indicated in **Figure R13b**, whose CBM is lower than CBM of 2L-WS₂ by around 0.13 eV, as indicated in Ref [3]), which is expected to show a smaller dipole moment than that of the heterostructure formed by 3L-InSe. However, during our experiment, we unfortunately failed to find

the 2L-InSe (also, the InSe layer with very thin thickness is not as detectable as WS₂ in the experiment [17]) with large enough area to form the heterostructure. Although the 2L-InSe sample is absent, the heterostructures composed of InSe, with layer numbers ranging from 3L to 6L, sufficiently demonstrate a trend of the increasing dipole moment with the increasing layer number.

Compared with the 2L InSe, monolayer (1L) InSe shows a dramatic increase (> 0.5 eV, according to the calculated results in ref [18, 19]) in the bandgap. CBM of 1L-InSe would be higher than that of 2L-WS₂ (**Figure R13b**), preventing the formation of type-II alignment.

Figure R13. Schematic band structures for heterostructure composed of multilayer InSe and WS₂ (a). Comparison of band structures of 2L-InSe/2L-WS₂ with that of 1L-InSe/2L-WS₂ (b).

1L WS₂ is the direct bandgap semiconductor with VBM resides at K point (as indicated in **Figure R14a**) instead of Γ point, which consequently doesn't align with our purpose. In comparison, 2L-(>2 L) WS₂ meets this requirement with its VBM resides at Γ point (**Figure R14b**).

Figure R14. Band structures for 1L-WS₂ (a) and 2L-WS₂ (b) with images cited from Figure 3 and Figure 4 in Ref [20] respectively.

Comment 2. The results suggest that thicker heterostructures yield larger dipole moments, with the maximum observed in 3L WS₂/6L InSe. Does this trend continue beyond 6L InSe or 3L WS₂? Is there a theoretical or experimental upper limit to the dipole moment in such systems? A discussion on this would provide deeper insight.

Response: We sincerely appreciate the reviewer for this insightful suggestion, which has facilitated a deeper discussion on interlayer excitons in multilayer heterostructures. As shown in **Figure R15** (the same as **Figure R11**), the increase in dipole moment with the rising number of InSe or WS₂ layers is not saturated. There are primarily two factors that will limit the maximum dipole moment we can achieve:

1. Disruption of Type-II Alignment with Increasing InSe and WS₂ Layers

As shown in **Figure R16a**, the energy difference between the VBM of InSe and that of WS₂ decreases from 0.39 eV to 0.18 eV as the number of InSe layers increases from 2L to 5L (according to Ref. [3]). Experimental data (**Figure R16b**, directly cited from **Figure 3a** in the reference [npj 2D Mater Appl 8, 12 (2024)]) indicates that this energy difference approaches zero as the number of InSe layers approaches 18, rendering the type-II alignment invalid. Consequently, we can roughly estimate the upper limit of the dipole moment at 18L InSe as $p_{largest} = (2.68 \text{ enm}) + (0.3 \text{ enm} \times 12) = 6.28 \text{ enm}$. Here, 2.68 enm is the dipole moment for the 2L/6L heterostructure, and we estimate that each additional layer increases the total dipole moment by 0.3 enm (based on the increase from 2L/5L to 2L/6L).

A similar analysis can be applied to WS₂. As indicated in **Figure R16c**, the CBM of InSe and WS₂ approaches each other with an increasing number of WS₂ layers. However, the variation in the CBM of WS₂ with layer number is smaller (as seen in Peak I of **Figure R16d**, which is directly cited from **Figure 3c** in [Sci Rep 3, 1608 (2013)]) compared to the trend observed in InSe. Increasing the WS₂ layers to bulk would result in a CBM of WS₂ that is very close to that of InSe, rather than surpassing it, which would compete with interlayer exciton transitions. However, due to the lack of further band structure data for various layers of WS₂, the

estimation of the upper limits of the dipole moment with increasing WS₂ layers remains uncertain.

2. Binding energy:

The interaction between the electron and hole can be described by the Coulomb potential, which is proportional to $\frac{1}{\epsilon_r d}$ where ϵ_r is the relative permittivity of heterostructure (approximately treated as constant across different layer combinations) and d is the dipole size. Consequently, a larger dipole size results in weaker Coulomb attraction, thereby reducing the binding energy of the exciton.

In a previous report (supplementary materials in Ref. [3]), the binding energy for the 2L/3L heterostructure was estimated to be less than 20 meV. Given that the binding energy is proportional to $E_b \propto \frac{1}{d}$ and that $d=1.35$ nm for the 2L/3L combination, and considering that the binding energy must exceed the thermal energy of the environment (i.e., $k_B T \approx 7$ meV for $T=77$ K), we estimate the maximum dipole size to be around 3.85 nm. However, the limits set by binding energy can increase with a decrease in temperature, as this leads to a reduction in the thermal energy from the environment.

Revisions: The related discussion on the upper limits of the dipole moment by layer-engineering in the revised manuscript is attached:

“Last but not least, while the layer configuration up to the 3L/6L combination exhibits the largest dipole moment observed in our experiments, it is clear that the upper limits have not yet been reached (see **Supplementary Figure S25**). Two factors could lead to the disappearance of interlayer excitons (IXs) with increasing layer number. First, band structure alignment plays a crucial role. As the layer number increases, the type-II band structures (illustrated in **Figure 1b**) may become invalid, resulting in interlayer-intralayer exciton transitions. Second, binding energy is another critical factor. As the dipole size increases, the binding energy correspondingly decreases due to reduced Coulomb attraction. When the binding energy falls below the thermal energy of the environment, IXs can no longer be observed. Detailed analyses can be found in **Supplementary Note S3**, which provides upper limits for the dipole moment: approximately 6.28 nm due to the limitations by band structure alignment and about 3.85 nm due to the limitations by binding energy at 77 K. The latter estimation can be increased with the decreased temperature.”

Figure R15. The evolution of dipole moment with different layer combinations.

Figure R16. (a) Evolution of band structures of 2L-WS₂/NL-InSe heterostructure with the increasing layer number of InSe. (b) Measured PL spectra for InSe of various layer number (directly cited from Figure 3a in [*npj 2D Mater Appl* **8**, 12 (2024)].). (c) Evolution of band structures of NL-WS₂/2L-InSe heterostructure with the increasing layer number of WS₂. (d) The PL peak positions for WS₂ of various layer number (directly cited from Figure 3c in [*Sci Rep* **3**, 1608 (2013)].).

Comments 3. While PL spectroscopy effectively probes interlayer exciton energies, sometime reflectance measurements could offer additional insights—particularly in distinguishing intralayer exciton contributions and hybrid interlayer exciton states. Including such data would allow direct comparison with prior studies and enhance the manuscript’s robustness.

Response: We thank reviewer for this question and we agree on that that reflection (or transmission/absorption) measurements would offer additional insights. People did find the interlayer excitons and their hybridization with intralayer excitons in homostructure through reflection measurements, such as bilayer MoS₂ or trilayer WSe₂ (for example, in Ref [7, 8]). Compared with the IXs in homostructure where hole is distributed across the whole layers (**Figure R17b**), the electron and hole of IXs in heterostructures are separated in different layers (**Figure R17a**), leading to the small oscillator strength (which is estimated of 10⁻³ to 10⁻⁴ smaller than that of intralayer exciton, according to Ref [6]) owing to the reduced electron-hole wave function overlap. As a consequence, for IXs in heterostructure people (including us) majorly adopted PL measurements for the characterization.

We measured the reflection spectrum for an representative sample with the 2L/5L configuration. As indicated in **Figure R18**, the reflection spectrum doesn’t show the visible signals at around IX resonance (which is reflected in corresponding PL spectrum). In fact, only the intralayer A exciton of WS₂ has shown the pronounced dip in the reflection spectrum (the intralayer exciton in InSe has a out-of-plane dipole moment [12] which is thus not detectable under normal reflection measurement).

Revisions: Figure R18 was added as Supplementary Figure S28 in the revised supplementary materials.

Figure R17. Schematic comparison of the formation of IXs in heterostructures and homostructures.

Figure R18. Measured reflection (orange curve) and PL (green curve) spectra of a representative 2L/5L heterostructure.

Comment 4. The manuscript contains numerous spelling errors (e.g., References [5, 9, 13, 14, 19, 30, 31...]), suggesting insufficient attention to detail. Journal name formatting is inconsistent (e.g., “Nature Communications”, “Nat Commun”, “Nat Photonics”). The references should follow a uniform style (preferably abbreviated journal names, e.g., Nat. Commun., Nat. Photon.).

Response: We thank the reviewer for pointing out these issues. We carefully checked the manuscript and corrected all typos we can find in the revised manuscript, including the spelling errors and inconsistent formatting.

References:

- [1] Sun, Z., Ciarrocchi, A., Tagarelli, F. et al. Excitonic transport driven by repulsive dipolar interaction in a van der Waals heterostructure. *Nat. Photon.* **16**, 79–85 (2022).
- [2] Wietek E, Florian M, Göser J, et al. Nonlinear and negative effective diffusivity of interlayer excitons in moiré-free heterobilayers. *Phys. Rev. Lett.*, **132**, 016202 (2024).
- [3] Ubrig, N. *et al.* Design of van der Waals interfaces for broad-spectrum optoelectronics. *Nat Mater* **19**, 299–304, (2020).
- [4] Yu, L., Pistunova, K., Hu, J. *et al.* Observation of quadrupolar and dipolar excitons in a semiconductor heterotrilinear. *Nat. Mater.* **22**, 1485–1491 (2023).

- [5] Sun, Z., Ciarrocchi, A., Tagarelli, F. *et al.* Excitonic transport driven by repulsive dipolar interaction in a van der Waals heterostructure. *Nat. Photon.* **16**, 79–85 (2022).
- [6] Barré, E. *et al.* Optical absorption of interlayer excitons in transition-metal dichalcogenide heterostructures. *Science* **376**, 406–410 (2022).
- [7] Leisgang, N., Shree, S., Paradisanos, I. *et al.* Giant Stark splitting of an exciton in bilayer MoS₂. *Nat. Nanotechnol.* **15**, 901–907 (2020).
- [8] Zhang, Y., Xiao, C., Ovchinnikov, D. *et al.* Every-other-layer dipolar excitons in a spin-valley locked superlattice. *Nat. Nanotechnol.* **18**, 501–506 (2023).
- [9] Karni O, Barré E, Lau S C, et al. Infrared interlayer exciton emission in MoS₂/WSe₂ heterostructures. *Phys. Rev. Lett.* **123**, 247402 (2019).
- [10] Tan Q, Rasmita A, Li S, et al. Layer-engineered interlayer excitons. *Science Advances*, **7**(30): eabh0863 (2021).
- [11] Lau K W, Cocchi C, Draxl C. Electronic and optical excitations of two-dimensional ZrS₂ and HfS₂ and their heterostructure. *Physical Review Materials*, **3** 074001 (2019).
- [12] Brotons-Gisbert, M., Proux, R., Picard, R. *et al.* Out-of-plane orientation of luminescent excitons in two-dimensional indium selenide. *Nat Commun* **10**, 3913 (2019).
- [13] Molina-Sánchez A, Sangalli D, Hummer K, et al. Effect of spin-orbit interaction on the optical spectra of single-layer, double-layer, and bulk MoS₂. *Phy. Rev. B*, **88**, 045412 (2013).
- [14] Torun E, Miranda H P C, Molina-Sánchez A, et al. Interlayer and intralayer excitons in MoS₂/WS₂ and MoSe₂/WSe₂ heterobilayers. *Phy. Rev. B*, **97**, 245427 (2018).
- [15] Wu K, Zhong H, Guo Q, et al. Identification of twist-angle-dependent excitons in WS₂/WSe₂ heterobilayers. *National Science Review*, **9**, nwab135 (2022).
- [16] Ciarrocchi, A., Unuchek, D., Avsar, A. *et al.* Polarization switching and electrical control of interlayer excitons in two-dimensional van der Waals heterostructures. *Nature Photon* **13**, 131–136 (2019).
- [17] Yi-Ying Lu *et al.*, Optical determination of layered-materials InSe thickness via RGB contrast method

and regression analysis. *Nanotechnology* **33** 485702 (2022).

[18] Paylaga N T, Chou C T, Lin C C, et al. Monolayer indium selenide: an indirect bandgap material exhibits efficient brightening of dark excitons. *npj 2D Materials and Applications*, **8**, 12 (2024).

[19] Sang D K, Wang H, Qiu M, et al. Two dimensional β -InSe with layer-dependent properties: band alignment, work function and optical properties. *Nanomaterials*, **9**, 82 (2019).

[20] Roy S, Bermel P. Electronic and optical properties of ultra-thin 2D tungsten disulfide for photovoltaic applications. *Solar energy materials and solar cells*, **174**, 370-379 (2018).

Response letter for the manuscript (NCOMMS-25-35571) entitled “Observation of Giant Dipole Moments of Interlayer Excitons via Layer Engineering”

Reviewer #1

Comment 1: I thank the authors for the reply. I appreciate that the authors clarify the interlayer to intralayer character transition of this specific heterostructure. This does not remove my concern about the impact of achieving large dipole but only within very small field, i.e. only limited ΔE . Nevertheless I acknowledge the novelty of this feature in its own context compared to K valley exciton in MX₂. The added power dependent PL makes sense but not unexpected. The authors stated that their 2L/3L device can not stand a field beyond 0.08 V/nm. which seems like a very low threshold. A second device or some clarification would be encouraged.

Our response: We thank the Reviewer for this question. The maximum electric field that can be applied to the device is determined by the limits of the electric field that h-BN can withstand. These limits can be attributed to defects, such as vacancies or impurities, which create localized states within the bandgap that facilitate tunneling and conduction. **Figure R1b** illustrates the evolution of leakage current with the applied electric field for the 2L-WS₂/3L-InSe device (**Figure R1a**). When the electric field exceeds 70 mV/nm, the leakage current across the device increases dramatically. Consequently, we refrain from further increasing the field to avoid damaging our devices.

The limits of the applied field vary slightly among different devices, ranging from approximately 80 mV/nm to 120 mV/nm, due to the device variations (please check the **Figure S19** in the Supplementary Material). In comparison to previous studies on interlayer excitons (see **Figure R2**, refs. [1-3]), the electric field ranges applicable to our heterostructures are similar, albeit slightly smaller. This range provides a sufficient linear Stark shift for extracting dipole moments across various layer combinations.

Figure R1. (a) Schematic illustration of experimental setup of 2L-WS₂/3L-InSe heterostructure encapsulated between two hBN layers. The leakage current (orange arrow) is detected when the vertical electric field is increased. (b) The current versus the applied electric field.

Figure R2. Cited images from the previous publishes. (a-c) from ref. [1], [2], and [3] separately.

Comment 2: Note that the authors reply that they layer engineered the IX dipole in heterostructure for the first time, but I disagree with such claim. Fig 6c in supp info of Nat. Mater. 19, 630–636 (2020) showed clear dipole tuning of K valley IX with various component thickness. Similarly see also Fig S7 and S8 in Ref.19 indicating similar effect on K valley IX. Various papers with heterobilayer separated by BN spacer in principle is also a type of dipole-layer engineering.

Our response: We appreciate the Reviewer for bringing this issue to our attention. The primary contribution of our work is a systematic study of the layer-engineered dipole moment of interlayer excitons, as demonstrated by the linear Stark effects. Additionally, the dipole-dipole interaction is enhanced with the increasing dipole moment, and the underlying mechanism is revealed through first-principles calculations. We modified our expressions in the revised manuscript with the related sentences listed below:

“In this study, we systematically investigate the variation of interlayer dipole moments in multilayer heterostructures using the quantum-confined Stark effect.”

Comment 3: Overall in my opinion the authors did not provide exceptional critical new results in their revision, but I acknowledge the findings provide new insight for dipole engineering and interlayer-intralayer transition in the context of Γ -related excitons. Authors reply to all the technical questions from other referees in details. I can suggest acceptance of the paper after addressing the comments above.

Our response: As Reviewer mentioned, the major contributions of our work are providing the new insight of the layer-engineering dipole moment and observation of the giant dipole moment in the experiment. We thank Reviewer for the recognition of our work and appreciate the suggestions and comments by the Reviewer.

The authors have successfully addressed my concerns. The paper is in good shape now and can be accepted.

Our response: We thank Reviewer for the insightful suggestions and recognition of our work.

Reviewer #3

The authors addressed all of my questions thoroughly. I recommend that this manuscript be published in Nature Communications.

Our response: We appreciate the comments and suggestions by the Reviewer, and thank for the recognition of our work.

References:

- [1] Ciarrocchi, A., Unuchek, D., Avsar, A. *et al.* Polarization switching and electrical control of interlayer excitons in two-dimensional van der Waals heterostructures. *Nature Photon* **13**, 131–136 (2019).
- [2] Meng, Y., Ma, L., Yan, L. *et al.* Strong-interaction-driven quadrupolar-to-dipolar exciton transitions in a trilayer moiré superlattice. *Nat. Photon.* (2025).
- [3] Yu, L., Pistunova, K., Hu, J. *et al.* Observation of quadrupolar and dipolar excitons in a semiconductor heterotrilaier. *Nat. Mater.* **22**, 1485 – 1491 (2023).